# Understanding water behaviour on 2D material interfaces through single-molecule motion on h-BN and graphene

Phillip Seiler [1,5], Anthony J. R. Payne[2,3,5], Neubi F. Xavier Jr [2],
Louie Slocombe [2,4], Marco Sacchi [2] ✉ & Anton Tamtögl [1] ✉

Understanding water behaviour on 2D materials is crucial for applications in sensing, microfluidics, and tribology. While graphene-water interactions are well studied, water on hexagonal boron nitride (h-BN) remains largely unexplored. Despite its structural similarity to graphene, h-BN possesses polar B-N bonds that give rise to distinct electronic and chemical properties. Most previous studies have also focused on multilayer water, leaving single-molecule dynamics poorly understood. Here we show how individual water molecules diffuse on h-BN compared to graphene using helium spin-echo spectroscopy and ab initio calculations. On h-BN/Ni, water exhibits coupled rotational-translational motion, in contrast to the discrete hopping observed on graphene. Water molecules rotate freely around their centre of mass, and although binding energies are similar on both materials, the activation energy for water dynamics on h-BN is 2.5 times lower than on graphene. These dynamics, which classical models fail to capture, highlight the fundamentally different nature of water transport on polar 2D surfaces. We further demonstrate that the supporting substrate strongly influences water friction, with h-BN/Ni showing markedly lower friction than graphene/Ni, opposite to the behaviour of free-standing layers. These findings challenge assumptions and offer insights for designing microfluidic devices requiring precise control of water mobility.

Graphene and hexagonal boron nitride (h-BN) are among the most important and well-investigated 2D materials and find applications in fields as varied as coatings, constructions, nanotechnology, sensors, biomedics and microfluidics[1-3]. Graphene and h-BN both have a honeycomb structure with very similar lattice constants but completely different electronic properties. Isolated and pristine graphene is a zero-band gap material, while h-BN is an electric insulator with a band gap close to 6 eV[4]. Both materials have extraordinary mechanical resistance and thermal conductivity, which makes them desirable for chemical applications such as catalysis and filtration[1,5,6]. Water-surface interactions are ubiquitous in these processes, determining structural, dynamic, and chemical properties in applications, such as water purification[2,7,8], drug delivery in aqueous media[9-11], and hydrogen storage[12,13].

Understanding the molecular basis of water-surface interactions is thus crucial for both technological advancements and fundamental physics[14,15], providing valuable insight into the nature of water/2D material interfaces, and opportunities for the design of advanced

[1]Institute of Experimental Physics, Graz University of Technology, Graz, Austria. [2]School of Chemistry and Chemical Engineering, University of Surrey, Guildford, UK. [3]School of Chemistry and Chemical Engineering, University of Southampton, Southampton, UK. [4]BEYOND Center for Fundamental Concepts in Science, Arizona State University, Tempe, AZ, USA. [5]These authors contributed equally: Phillip Seiler, Anthony J. R. Payne. ✉e-mail: m.sacchi@surrey.ac.uk; tamtoegl@tugraz.at

materials by tuning their nanoscopic properties[16–19]. However, while properties such as wettability and ice nucleation at the macroscopic level are well understood[20], measurements at the atomic or molecular level are scarce and focused on metal substrates[21–24] or metal-supported thin ionic films such as NaCl[25]. In addition, previous ab initio computational studies on the interaction of water with 2D material surfaces have primarily examined the adsorption of water, with dynamic behaviour often modelled using classical molecular dynamics simulations[26]. On the one hand, 2D materials can be used as protective coatings of metal and semiconducting surfaces, to reduce corrosion and inhibit interdiffusion, wetting and icing[8,27–29]. On the other hand, defect-free 2D material synthesis is best achieved via low-pressure chemical vapour deposition (CVD), and here we have specifically chosen Ni(111) as a substrate in the CVD growth because of the perfect lattice match with both graphene and h-BN.

In the context of water interactions with 2D materials, friction plays a crucial role and understanding the relationship between the energy landscape and surface vibrations (i.e., phononic or "mechanical" friction) remains a central challenge. Recent theory suggests a quantum contribution to friction at the solid/liquid interface, explaining the anomalously high friction of water on graphite compared to graphene[30], a phenomenon also linked to electron cooling at

the water/graphene interface[31]. Since h-BN is an insulator, unlike graphene, water dynamics on h-BN are especially interesting due to potentially different electronic couplings. Tocci et al. predicted higher macroscopic friction on h-BN via classical and ab initio simulations[32,33], and recent work shows water's vibrational modes influence interfacial friction with graphene[34]. However, many such studies rely on classical force fields[35]. Here, we instead use an ab initio description of a single $H_2O$ molecule, together with experimental data, to go beyond classical models.

Experimental investigations of water/2D interfaces are limited by the challenges of imaging picosecond and sub-angstrom dynamics[36]. The high proton mobility and delocalisation in $H_2O$, even at low temperatures, hinder atomic-level characterisation[37]. Furthermore, water is easily damaged by electrons and high-energy photons[22,38,39]. Helium spin-echo (HeSE), with its picosecond resolution and use of non-destructive, low-energy He atoms, circumvents these issues[40–43]. A previously reported HeSE study of water motion on graphene[44] illustrated the stark contrast between water diffusion on weakly interacting 2D materials such as graphene[26,45] and metal substrates, where water is bound much stronger and further influenced by hydrogen bonding between the molecules[23,39,46]. To establish whether this is a general trend on 2D materials, further studies extending to, e.g., h-BN, are required. Moreover, the mentioned graphene study did neither consider the role of additional molecular degrees of freedom, nor the influence of single-molecular friction on water mobility for a supported 2D material[26].

In the present study, we demonstrate that for $H_2O$ diffusion on epitaxial graphene and h-BN, the single-molecule perspective is highly complex and incompatible with both a single-point particle diffusion model and with liquid water behaviour. Using the HeSE technique, illustrated in Fig. 1a, we measure surface correlations in the water motion. In contrast to water motion over graphene[44], single molecules can easily reorient on h-BN, and their motion can no longer be treated as a series of jump-like motions by point-like particles occupying equivalent adsorption sites. By analysing the dephasing rates in correlation measurements, we find that despite a water adsorption energy similar to that on graphene (within 16%), the activation energy on h-BN is much smaller than on graphene. The dynamics involve a fast rearrangement of the water molecule orientation during diffusion, in which the O-H bonds precess around the water centre of mass during translational motion. We rationalise these findings in the context of the multidimensional potential energy surface (PES) during diffusion and molecular friction on h-BN/Ni(111). With ab initio calculations, including a detailed quantum chemical study using density functional theory (DFT) and ab initio molecular dynamics (AIMD) simulations, we show how water behaves differently on h-BN than on graphene and establish the influence of the supporting metal substrate on the friction coefficient. The concept of friction as a single parameter/average over a microscopic ensemble is typically used at the macroscopic level, e.g., for liquid/solid interfaces, as mentioned above[32,33]. However, we illustrate that atomistic details such as the potential energy corrugation and the changes in the electronic structure induced by the supporting substrate are important, e.g., the behaviour of water on h-BN and graphene is reversed if the support is explicitly included in the theoretical model.

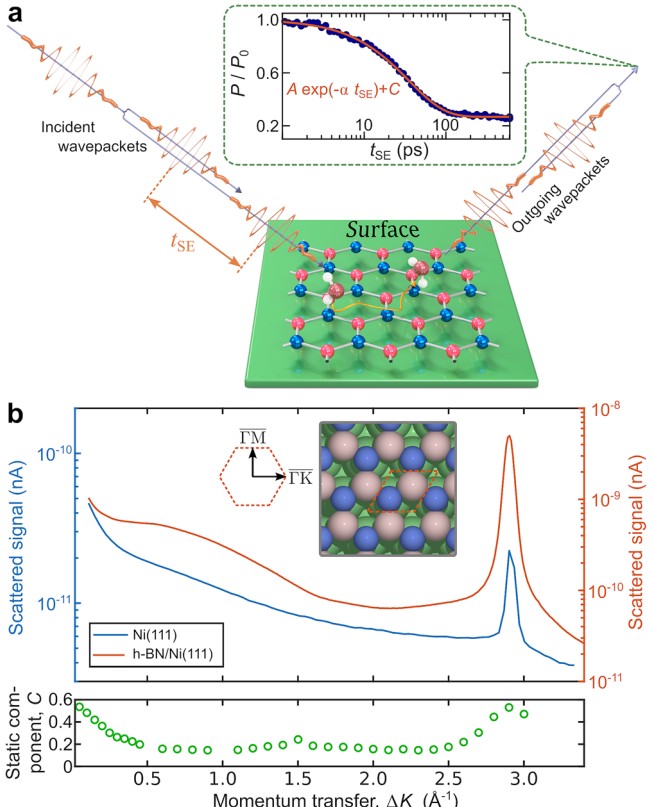

**Fig. 1 | Measurement of single-molecule water diffusion on h-BN. a** Illustration of the HeSE method: Two wavepackets scatter from the surface with a time difference $t_{SE}$, allowing the motion of molecules on the surface to be determined by the loss of correlation, which is measured through the polarisation of the beam. The inset shows a typical measurement of the diffusion of water on h-BN ($T = 120$ K, $\Delta K = 0.2$ Å$^{-1}$). The reduction in surface correlation with increasing spin-echo time follows a single exponential decay (solid line), characterised by the dephasing rate, $\alpha$. **b** A one-dimensional diffraction scan illustrates the epitaxial growth with the same symmetry as that of the pristine Ni substrate. The greater intensity of the h-BN peak compared to that of the substrate peak indicates the stronger corrugation of h-BN. The bottom panel shows the elastic component after exposing the surface to water ($C$ in (1)), illustrating that no ordered superstructure due to the adsorbed water is present. Source data are provided as a Source data file.

## Results

### Multidimensional energy landscape of $H_2O$ on h-BN

We first address the deficiency of single-molecule studies for water on h-BN and other supported 2D materials by establishing the energy landscape for water binding and adsorption on h-BN, including the complete system with the corresponding metal substrate. With the exception of ice structure studies on metal-supported h-BN forming a Moiré superlattice[47], where similarities to water on a graphene Moiré superlattice[48,49] and on functionalised graphite[50] were found, no single-

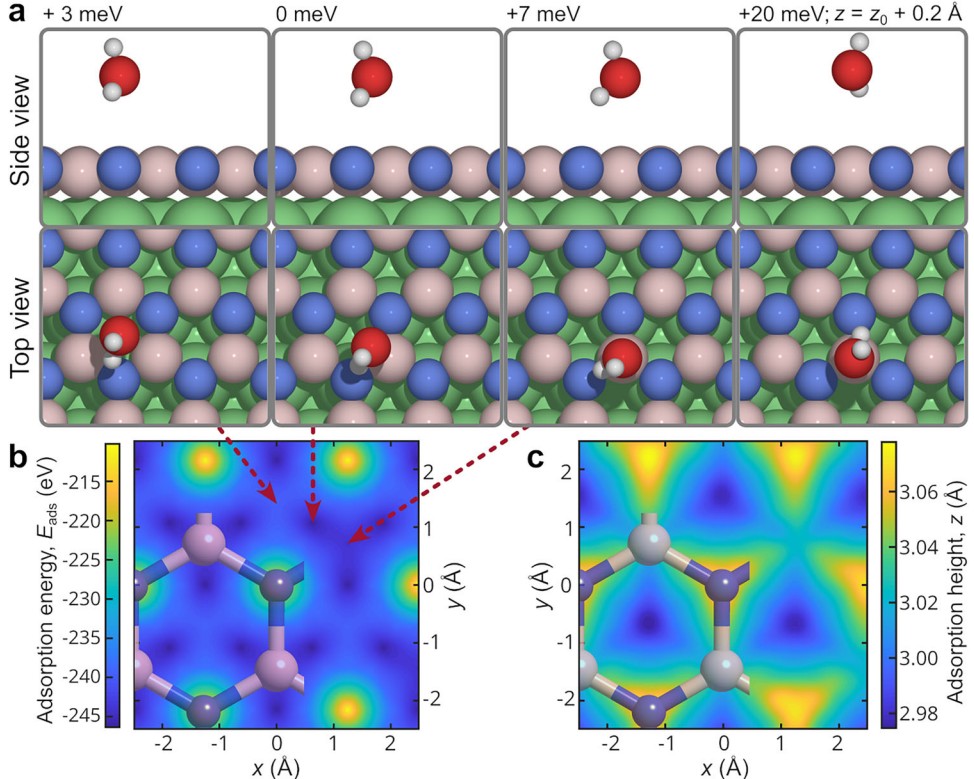

**Fig. 2 | Adsorption energy landscape from ab initio methods.** An investigation of the adsorption energy landscape by DFT reveals a weakly corrugated potential energy surface (PES). The adsorption geometries shown in (**a**) illustrate that water favours adsorption near a boron atom (pastel pink), with the hydrogen atoms pointing towards a nitrogen atom (blue) due to the weak intermolecular bond between the oxygen and the partially positively charged boron atom (see text). The PES as a function of $x$ and $y$ in **b** is characterised by small energy differences between the sites, with the only exception being the nitrogen atom site. The PES further presents a rather weak $z$-dependence of the adsorption sites, as shown in **c**, with the rightmost panel in **a** illustrating the geometry for a $z$ distance of 0.2 Å above the minimum energy site. Source data are provided as a Source data file.

molecular studies on h-BN/Ni(111) are available. Thus, to get a full picture of the adsorption behaviour of water on h-BN/Ni(111), we investigated the system both experimentally by conducting extensive adsorption and desorption measurements and theoretically using detailed vdW-corrected DFT calculations.

In the experiments, the Ni(111) substrate was first prepared under ultrahigh vacuum (UHV) conditions[47], and h-BN was grown following a chemical vapour deposition (CVD) process according to Auwärter et al.[51] (see Methods-Experiment and sample preparation and Supplementary Note 1.1). To verify the surface quality of the h-BN overlayer on top, we performed diffraction scans in the high-symmetry $\overline{\Gamma}\overline{M}$ orientation before and immediately after the CVD growth (see Fig. 1). The position of the first order diffraction peak at $\Delta K \approx 2.9$ Å$^{-1}$ clearly shows that the h-BN layer exhibits the same periodicity as the Ni(111) surface and that a clean, ordered overlayer of h-BN has been formed. We then started to determine the conditions under which individual water molecules stick on the surface and remain mobile, by conducting extensive adsorption and desorption experiments on the h-BN/Ni surface.

Continuous dosing with water below 120 K (Supplementary Note 1.3 and Supplementary Fig. 2) results in the formation of an amorphous layer of solid water covering the entire surface[44], while increasing the temperature to 135 K prevents full surface coverage, indicating hydrophobic behaviour and island formation on h-BN/Ni, similar to water adsorption on graphene/Ni[44]. As such it does not exclude the formation of clusters or islands; however, once small aggregates such as dimers have formed, they are unlikely to dissociate, and, as discussed below, exhibit significantly reduced diffusivity. The elastic component, from the dynamics measurements shown in the

bottom panel of Fig. 1b, confirms that no ordered H$_2$O structure is present for the experimental conditions of the dynamics measurements[52]. By performing thermal desorption measurements (Supplementary Fig. 4), in which the He reflectivity is measured during successive surface heating, we estimate the desorption energy to $\approx (0.53 \pm 0.06)$ eV, which is very similar to the experimentally determined 0.52 eV for water on graphene/Ni(111)[44].

To obtain a better understanding of the water-surface interaction of a single molecule, we performed a detailed quantum chemical study of the geometries and adsorption energies ($E_{ads}$) of H$_2$O on h-BN/Ni(111) (see Methods section Theoretical methods). Initially, seven high-symmetry sites were chosen (Supplementary Fig. 6) to construct a PES as a function of the oxygen position. The oxygen coordinates were fixed at a given position, and the hydrogen atoms were allowed to relax. The molecular orientation was optimised to find the most favourable conformation for each site (the more negative $E_{ads}$, the stronger the adsorption). Further optimisations in which the oxygen atom was allowed to fully relax were included to find the global minimum (the most stable configuration of the adsorbate-surface system). The energy differences between different adsorption sites indicate a weakly corrugated PES (Fig. 2b), where minor changes in the adsorption height of the water molecule result in minimal energy variation (Fig. 2c).

As shown in Fig. 2a, H$_2$O physisorbs near a boron atom and exhibits an $E_{ads}$ of $-0.25$ eV (compared to an $E_{ads}$ of $-0.21$ eV for graphene/Ni, as shown in Supplementary Fig. 6). This physisorption site is favoured due to the interaction between the oxygen lone pair and the partially positively charged surface boron atom, forming a weak intermolecular bond. Additionally, one of the hydrogen atoms in the water molecule and the nearest surface nitrogen atom engage in

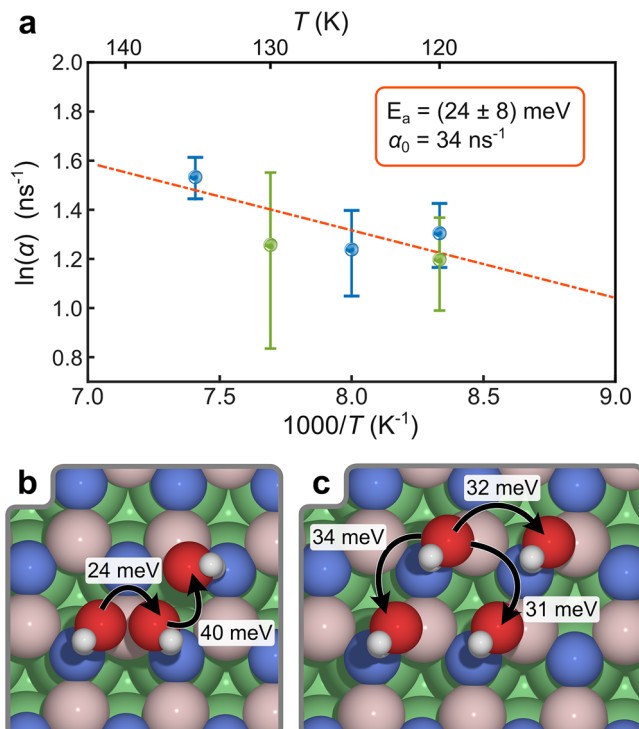

**Fig. 3 | Small activation energies for water dynamics on h-BN/Ni(111).**
**a** Temperature-dependent measurements at a constant momentum transfer of $\Delta K = 0.6\ \text{Å}^{-1}$ along both high-symmetry orientations $\overline{\Gamma M}$ (blue data points) and $\overline{\Gamma K}$ (green data points) show an extremely low activation energy of $E_a = 24$ meV. Error bars are the corresponding confidence bounds (1$\sigma$) of the exponential fits (see text). **b, c** Pathways showing the migration of $H_2O$ between sites on h-BN/Ni(111) along with the associated DFT-calculated transition state energy barriers, which are consistent with the experimentally determined barrier within the reported uncertainty. **b** illustrates translation coupled with rotation, while **c** demonstrates translation without rotation. The consistently low energy barriers along each pathway suggest ready accessibility, facilitating migration of water across the surface. Source data are provided as a Source data file.

is considered to be the most favourable in the context of free-standing h-BN.

**A low barrier for water dynamics on h-BN**
In the next step, we establish the energies which are necessary for the onset of single-molecular $H_2O$ dynamics on h-BN/Ni and compare those to the graphene/Ni system. To study the molecular motion of water on the h-BN/Ni(111) surface with a temporal sensitivity on the picosecond timescale, we performed HeSE experiments in the temperature range of 120–135 K at a $H_2O$ coverage between 0.12 and 0.20 monolayer (ML) (see Supplementary Note 1.2). At this coverage well below one monolayer, the effects of correlated motion are negligible, allowing us to study the single-molecule dynamics. During a single spin-echo measurement, the decay of the intermediate scattering function (ISF) versus the spin-echo time $t_{SE}$ provides a measure of the surface correlation and thus the timescale of motion at the length scale corresponding to the momentum transfer $\Delta K = |\Delta \mathbf{K}|$. By measuring the ISF over a range of temperatures and $\Delta K$ values, we obtain details about the diffusive motion and its length scales in real space. As illustrated in the inset of Fig. 1a, each measurement can be described by a single-exponential function:

$$I(\Delta K, t_{SE}) = A \exp\left[-\alpha(\Delta K)\, t_{SE}\right] + C(\Delta K)\ . \tag{1}$$

In Eq. (1), A is the amplitude at $t = 0$, the constant $C(\Delta K)$ represents the static level of the surface, i.e., the fraction of the signal that arises from immobile parts on the surface. The prefactor in the exponent $\alpha(\Delta K)$ is the so-called dephasing rate. The dephasing rate, in units of $s^{-1}$, measures how fast diffusion on the surface occurs. Its functional dependence on the momentum transfer $\Delta K$ parallel to the surface contains a variety of information and provides a signature of both the rate and mechanism of molecular motion.

We measured the temperature-dependence of $\alpha$ at a fixed coverage and a fixed momentum transfer $\Delta K = 0.6\ \text{Å}^{-1}$ for both the high-symmetry $\overline{\Gamma M}$ and $\overline{\Gamma K}$ directions over a temperature range of 120–135 K (see the Arrhenius plot Fig. 3a). For thermally activated motion, the rate is given by Arrhenius' law,

$$\alpha = \alpha_0 \cdot \exp\left[-E_a / (k_B \cdot T)\right], \tag{2}$$

where $k_B$ is the Boltzmann constant, $T$ is the surface temperature of the sample and $E_a$ is the activation energy for diffusion. The uncertainties in the data points are the corresponding confidence bounds (1$\sigma$) of the exponential fits. From the slope of the linear fit in Fig. 3a, we obtain an activation energy of $E_a = (24 \pm 8)$ meV. We note that due to the proximity of the diffusion and desorption regimes, the temperature range is constrained to 120–135 K, within which coverage remains stable. To reduce the confidence interval of the linear fit in Fig. 3a, we include data points for both the $\overline{\Gamma M}$ (blue) and $\overline{\Gamma K}$ (green) azimuths, assuming that diffusion occurs at a similar rate along both directions, as also supported by AIMD calculations.

The experimental value can then be compared to theoretical models of the reaction pathway from DFT calculations, where the minimum energy pathway for water migration connects the global minimum energy site (the N atom) to the nearest equivalent site on h-BN/Ni. Following an evaluation of the transition state pathways, the energetically most favourable pathways are shown in Fig. 3b, c. Fig. 3b illustrates a sequence where the water molecule undergoes a dynamic process involving translation and rotations. We note that such multistep diffusion pathways can, in principle, alter the Arrhenius prefactor through entropic contributions to the transition state, although such effects are not resolved in the experimental dephasing observable $\alpha(\Delta K)$, and in general both prefactor and activation energy may vary with temperature, as shown, e.g., in ref. 53 The molecule first crosses a boron site, then a hollow site, before ultimately rotating into an

hydrogen bonding, which is characterised by Hirshfeld charges of $+0.17\,e$ and $-0.16\,e$ on the boron and nitrogen atoms, respectively, further enhancing the stability of this configuration. The high electron density and localised bond population of each B-N bond render the "bridge" sites of h-BN unfavourable for $H_2O$ adsorption. Consequently, the oxygen atom tends to relax slightly away from the site directly above the bond.

To determine the effect of the supporting metal substrate, the adsorption energetics and geometries of $H_2O$ on freestanding h-BN were also calculated and analysed. Similar to that on the h-BN/Ni surface, our DFT calculations of water adsorption on a free-standing h-BN layer reveal the formation of a hydrogen bond between the water molecule and a surface nitrogen atom (see Supplementary Fig. 6). The hydrogen bond stabilises this configuration, resulting in an $E_{ads}$ of $-0.18$ eV, which indicates slightly weaker adsorption compared to h-BN/Ni. In the case of freestanding h-BN, the distal hydrogen atom is oriented towards another surface nitrogen atom, with a H-N distance of 3.5 Å: This arrangement maximises the long-range hydrogen bonding. The differences between adsorption sites, orientations and bonding interactions between the freestanding h-BN and h-BN/Ni can be attributed to differences in charge localisation and polarity. In the case of freestanding h-BN, the surface nitrogen exhibits a higher Hirshfeld charge of $+0.21\,e$ and $-0.21\,e$ on the boron and nitrogen atoms, respectively, indicating an increased hydrogen bonding strength. Thus, the adsorption site that maximises hydrogen bonding

equivalent position adjacent to another nitrogen atom. The calculated activation energy barriers for traversing the boron and hollow sites are 24 meV and 40 meV, respectively. Following this motion, a barrierless rotation around the nitrogen atom occurs, returning the water molecule to its original orientation. Furthermore, alternative paths involving translation without molecular rotation were explored, yielding different energy barriers depending on the traversed surface sites Fig. 3c. The hollow site exhibits the lowest activation energy barrier at 31 meV, followed by the boron site at 32 meV, with the nitrogen site presenting the highest energy barrier at 34 meV. In each case, the transition state structure closely aligns with the midpoint between each energy minimum.

The calculated activation energy barriers to rotation across a boron site (24 meV) and translation without rotation (32–34 meV) are consistent with the experimental measurements with an $E_a$ of $24 \pm 8$ meV. As the energy barriers for these pathways are considerably low, each pathway is expected to be readily accessible and can occur with minimal energy input at low temperatures. Repeated traversals along these pathways likely lead to the isolated water molecule undergoing a sequence of jumping and spinning motions across the surface. Apparently, such behaviour of a single water molecule has not been reported previously, while the terminology for the molecular degrees of freedom follows from thiophene diffusion[54].

For completeness, we also examined the possibility of dimer formation, dissociation, and diffusion, as detailed in Section 2.2 of the Supplementary Information and illustrated in Supplementary Figs. 7 and 8. While dimer formation is thermodynamically allowed under low-coverage conditions, the associated diffusion barriers are substantially higher than those for single-molecule motion. Specifically, calculated barriers for dimer translation and rotation range from 79 to 220 meV, in contrast to the 24 meV barrier observed experimentally for monomer diffusion. Moreover, dimer dissociation involves an energy cost of $\approx 220$ meV, indicating that dimers, once formed, are kinetically stable but effectively immobile under the experimental conditions. These findings support the conclusion that the observed dynamics predominantly arise from isolated water molecules and that single-molecule simulations provide the most accurate description of the experimental system.

To sum up, based on both experimental results and first-principles calculations, we conclude that the differences in $E_{\text{ads}}$ between various adsorption sites is very small, and the corrugation of the potential energy surface for water motion over h-BN/Ni (measured as the activation energy variation on the surface plane, Fig. 2) is particularly weak. Such a small barrier for the onset of motion could be easily overcome by tip interactions in STM measurements at cryogenic temperatures[55], whereas the low-energy He atoms do not interfere with the motion of $H_2O$. Most importantly, compared to single-molecule water diffusion on graphene/Ni, with an activation energy of $60 \pm 4$ meV[44], both the experimental and theoretical values for h-BN/Ni are considerably smaller by a factor of 2.5. These differences in activation energy suggest a significant distinctness of the molecule-surface interaction, including the underlying energy landscape and surface dynamical properties, which are explored in the following sections, including also the Molecular friction of water on graphene and h-BN.

## Walking motion of water on h-BN

Despite the similarity of the substrate structure and $H_2O$ adsorption energy on h-BN/Ni to those on graphene/Ni, we observe a pronounced difference in the multidimensional energy landscape and a low activation energy for molecular dynamics on h-BN/Ni(111). In the following, we address the details of $H_2O$ motion on h-BN/Ni, specifically considering the additional molecular degrees of freedom and the stark contrast in water motion on h-BN/Ni compared to graphene/Ni. The blue points in Fig. 4a show the variation in $\alpha(\Delta K)$ for water molecules at $T = 120$ K along the high-symmetry $\overline{\Gamma M}$ direction in reciprocal space.

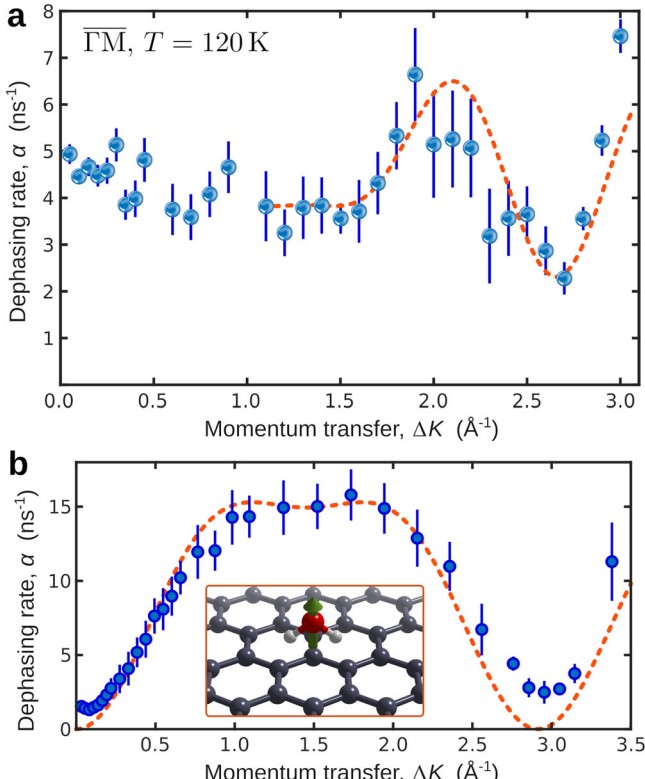

**Fig. 4 | Diffusion of water on hexagonal boron nitride. a** Momentum transfer dependence of the dephasing rate $\alpha(\Delta K)$ (blue dots) at $T = 120$ K, from which the diffusion mechanism of $H_2O$ on h-BN/Ni follows. An analytical model (red dash-dotted curve) shows that the motion contains a jump component for the translation of the molecules that follows the periodicity of the substrate. In addition, the motion is dominated by a strong normal component, which cannot be reproduced by the analytic model but is confirmed by ab initio calculations. The error bars correspond to the confidence bounds ($1\sigma$) in the determination of $\alpha$ from the measurements. **b** Single-molecule motion of $H_2O$ on graphene/Ni ($T = 125$ K) according to[44] for comparison, where the water dipole remains perpendicular to the substrate (see inset) and $H_2O$ moves through a series of discrete jumps (error bars as in a). Source data are provided as a Source data file.

For each data point, a single ISF was recorded and fitted to Eq. (1). We used an iterative routine to optimise the range for inclusion, in which the data show the expected deviation from an exponential for short times[44]. The error bars correspond to the confidence bounds ($1\sigma$) of the exponential fit. Our data exhibit several signatures suggesting that a complex interplay among several processes characterises the observed motion. A dip in the dephasing rate appears at $\Delta K \approx 2.8$ Å$^{-1}$, which is very close to the position in reciprocal space of the first-order diffraction peak (Fig. 1b).

By fitting the Chudley-Elliot (CE) model, which is normally used to describe hopping of a point-like particle between adsorption sites on a Bravais lattice,[44,54,56–60] to the data points with $\Delta K > 1$ Å$^{-1}$, as illustrated by the red dashed lines in Fig. 4, we can extract a set of "effective" mass transport coefficients. With this model, we obtain a residence time $\tau = 290$ ps, an average jump length $\langle l \rangle = 5.5$ Å and a diffusion constant $D = (0.26 \pm 0.5)$ nm ns$^{-1}$ at 120 K. However, the simple CE model for jump diffusion cannot reproduce our data in the entire $\Delta K$ range, especially for small values of $\Delta K < 1$ Å$^{-1}$. In the region of very low $\Delta K$ values, the contribution to the dephasing rate from motion parallel to the surface can be neglected, and we can isolate the effects of perpendicular motion, which makes this region particularly interesting for our analysis, whereas at higher $\Delta K$ values, parallel motion becomes evident, and both processes contribute to the signal[59,61]. We observe that the dephasing rate remains almost constant for $\Delta K < 1.5$ Å$^{-1}$ and,

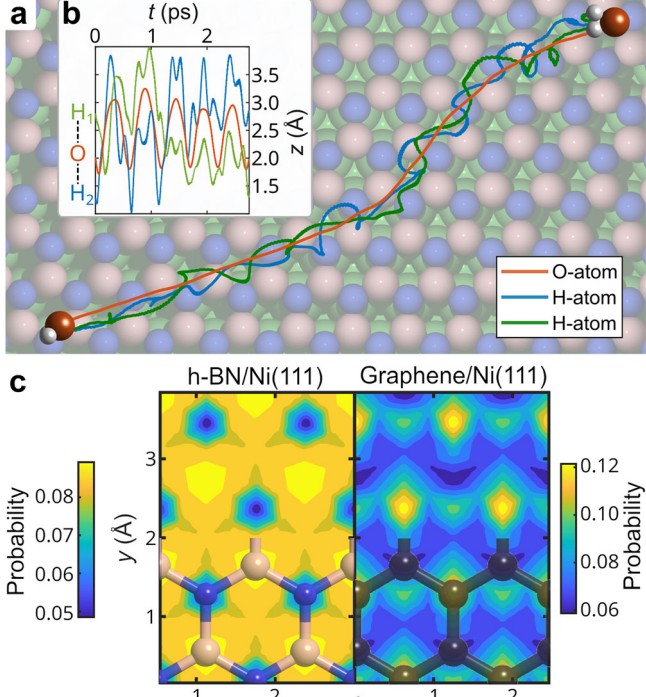

**Fig. 5 | Details of the molecular motion from ab initio methods. a** Typical trajectory for $H_2O$ on h-BN/Ni from AIMD simulations, illustrating that the hydrogen atoms precess around the oxygen atom along the trajectory, leading to a spinning motion perpendicular to the molecular direction of travel, similar to a corkscrew. The $z$-variation versus time plot in inset **b** shows that the position of the O-H bonds with respect to the oxygen atom can thus easily flip. **c** A stark contrast is observed when comparing the probabilities of water being located at a specific surface site during motion on h-BN/Ni (left panel) and graphene/Ni (right panel). For h-BN/Ni, $H_2O$ is hardly found at the N site, while the likelihoods for all the other sites are quite similar, leading to continuous motion. In contrast, on graphene/Ni, $H_2O$ motion most likely proceeds via jumps to adjacent equivalent sites, thus resulting in a more disconnected hopping motion. Source data are provided as a Source data file.

more importantly, remains at a nonzero constant value as $\Delta K$ approaches 0. Such a behaviour for $\Delta K \rightarrow 0$ is characteristic of confined motion perpendicular or parallel to the surface[26,54,59,61]. Confined motion can include several processes, such as rotation, flipping or spinning of the molecule with respect to its centre of mass[54]. The fact that our ISFs always decay to a finite value (see the static component in Fig. 1b) is also a clear indication of confined motion[59,61].

To investigate the dynamics in more detail and confirm the nature of the confined motion, we performed AIMD simulations at 150 K, which include all degrees of freedom of the adsorbed water molecule. Starting from the global minimum determined by DFT, as shown in Fig. 2, the trajectories and starting velocities were randomised in canonical simulations (NVT), see Methods-Theoretical methods, with an example trajectory plotted in Fig. 5a, while the real-space motion is illustrated by Supplementary Movies obtained from the AIMD simulations. Analysis of the oxygen atom path reveals a preference for trajectories that remain close to the nitrogen atoms, consistent with the PES and transition state energies discussed in the previous section. Furthermore, the $z$ coordinate of the oxygen atom (corresponding essentially to the height of the water molecule) oscillates as the water moves between the nearest neighbour minima (see the inset Fig. 5b). The observation of such a quasi-continuous, "walking" motion of water on h-BN/Ni(111), characterised by a combination of translational and rotational dynamics, shares conceptual similarities with prior studies of single-molecule water diffusion on NaCl(001) in refs. 25,62 : Water

on a NaCl bilayer on Ag(111) undergoes a rocking motion, where translations are coupled with rotation, driven by electrostatic interactions with $Na^+$ ions and hydrogen bonding with $Cl^-$ ions[25].

The regular oscillations indicate periodic surface features with consistent water-surface interactions and resulting forces. The movement of the hydrogen atoms, which precess around the oxygen trajectory, indicates a fast spinning motion of the molecule perpendicular to its direction of travel, meaning that the position of the O-H bonds with respect to the oxygen atom can easily flip (averaging 1.64 rotations per picosecond). Supplementary Movie 1 illustrates the mentioned trajectory, while Supplementary Movies 2 and 3 illustrate other trajectories, e.g., in the case of Movie 3 with a more confined motion. For the other movies, please refer to Data availability. These trajectories can also be used to analytically calculate the ISFs (see Supplementary Note 2.5). For vanishing $\Delta K$, the analytical ISFs exhibit a decay analogous to our experimental ISFs, the results confirm that the motion along the $z-$ component contributes to the constant offset of $\alpha(\Delta K)$. For comparison, Fig. 4b displays $\alpha(\Delta K)$ for graphene/Ni, which exhibits a negligible contribution from rotational motion relative to h-BN, and is instead dominated by repulsive intermolecular interactions. For a detailed analysis, see refs. 44,63 and the supporting information of ref. 44.

To better understand the nature of the observed dynamic motion, the spatial probability distribution of water was calculated on h-BN/Ni and graphene/Ni from AIMD trajectories. A comparison of the probability distributions in Fig. 5c indicates that the water motion in the two systems substantially differs. For the h-BN/Ni system, the probability distribution demonstrates notable uniformity among the sites, including the boron atom and hole sites and their interconnections, with the only exception being the nitrogen atom site. In contrast, the site probability for graphene/Ni as shown in the right panel of Fig. 5c is highly localised, leading to the discrete jumps observed in experiments (see Supplementary Note 2.3). The probabilities for h-BN/Ni coincide with the PES determined through assessment of the water $E_{ads}$, with higher probabilities at the preferred positions (Fig. 2), which favour close proximity to the boron atom and avoidance of the nitrogen atom.

The more uniform distribution of water on h-BN/Ni is also evident from the present experimental data, while for graphene/Ni, $H_2O$ most likely sits on one specific site, which requires the molecule to traverse a larger distance across the surface, resulting in point-like jumps between these sites according to the CE model, as has been previously reported[44]. A more extensive comparison which includes also the free-standing h-BN and graphene systems can be found in Supplementary Fig. 9. Notably, at a temperature of 150 K for h-BN/Ni, the probability of finding the water molecule in the conformation with the minimum energy is 5.8%, while there is a 90% probability of finding the molecule within one of the 48 lowest energy sites (out of 240). These states are distributed across the surface and follow a similar pattern to the AIMD probability distribution, as illustrated in Supplementary Fig. 10. At low temperatures, a variety of water conformations coexist across sites; therefore, water diffusion on h-BN/Ni is more complex than that described by the jump-diffusion model of water on single-crystal metal surfaces.

In conclusion, despite the similarity of the substrate structure and $H_2O$ adsorption energy on h-BN/Ni to those on graphene/Ni, both experiments and theory show completely different motions of single water molecules on the two surfaces. The diffusion of water on h-BN is slower than that on graphene ($D = 0.26$ nm² ns⁻¹ cf. $D = 0.32$ nm² ns⁻¹ extrapolated to 120 K), but the activation energy for molecular motion is negligible compared to that on graphene because of the simple rearrangement, i.e., rotational and flipping motion, of the molecule on the substrate. These findings are in line with the underlying PES, which is much more uniform for water on h-BN/Ni than for water on graphene/Ni. In the following, we further rationalise these findings by

illustrating the strongly divergent behaviour of the two systems, considering the atomic-scale friction.

## Molecular friction of water on graphene and h-BN

In the previous sections, we demonstrated how the nature of the 2D substrate and the additional molecular degrees of freedom significantly impact the mobility of water on 2D materials. We rationalise these findings in terms of molecular friction and illustrate the correlation between friction and both vibrational coupling between surface and adsorbate and the corrugation of the PES. Specifically, we show how both changes of the potential energy surface corrugation and the vibrational coupling, upon inclusion of the supporting metal substrate underneath the 2D materials, give rise to a completely different molecular friction when compared to the free-standing 2D materials. Therefore, we performed a series of AIMD simulations on both h-BN and graphene, with and without the presence of a supporting metal substrate.

We employed the microcanonical ensemble, considering a supercell of the corresponding system (see Methods - Theoretical methods), to determine the friction of water. The friction coefficient $\lambda$ for each MD simulation was calculated using the Green-Kubo (GK) relationship, which is defined as:[32,64]

$$\lambda_{GK} = \lim_{t \to \infty} \lambda_{GK}(t) \quad , \tag{3}$$

with,

$$\lambda_{GK}(t) = \frac{1}{2A \, k_B \, T} \int_0^t \langle \mathbf{F}(0) \, \mathbf{F}(t') \rangle \, dt' \quad , \tag{4}$$

where $F(t')$ is the lateral force acting on the sheet at a time $t'$ and $A$ is the interfacial lateral area. The $\lambda_{GK}(t)$ up to 2 ps were calculated by averaging results from AIMD simulations, each involving a single water molecule on the examined surfaces. Figure 6a shows the $\lambda_{GK}(t)$ for a single water molecule on free-standing graphene and h-BN, as well as a comparison with that for water on metal-supported h-BN and graphene. A single error bar is shown for the graphene/Ni system, which illustrates the statistical uncertainty and the degree of convergence when the simulation is extended to 3 ps; further details are provided in Supplementary Note 2.6.

Overall, the $\lambda_{GK}$ values observed for freestanding h-BN ($19.9 \cdot 10^6$ Nsm$^{-3}$) are considerably greater than those for freestanding graphene ($6.2 \cdot 10^6$ Nsm$^{-3}$), by a factor of $\approx 3.2$. These results are consistent with the MD results reported by Tocci et al.[32], where classical force fields were assumed for MD simulations of systems containing $\approx 400$ water molecules and revealed that h-BN has a $\lambda_{GK}$ that is a factor of 3.1 greater than that of graphene (with values of $30 \cdot 10^4$ Nsm$^{-3}$ and $9.6 \cdot 10^4$ Nsm$^{-3}$ for h-BN and graphene, respectively). Following the same methodology for a water monomer on the Ni-supported 2D materials, we determined the molecular friction of the systems, with $\lambda_{GK}$ for graphene/Ni ($9.5 \cdot 10^7$ Nsm$^{-3}$) being 7.9 times greater than $\lambda_{GK}$ for h-BN/Ni ($1.2 \cdot 10^7$ Nsm$^{-3}$) throughout the simulation time of 2 ps. Thus, as illustrated in Fig. 6a, by including the substrate (Ni), we observe a major change in friction, and the relative relationship between the $\lambda_{GK}$ of h-BN and graphene is reversed, i.e., the friction of water on supported graphene is much greater than on supported h-BN.

Interestingly, in a previous study of water on NaCl bilayers, the role of the underlying substrate was also highlighted - specifically, the Ag(111) substrate was found to significantly affect water adsorption on NaCl/Ag(111) compared to bulk NaCl[25]. Although the study did not address friction, it clearly showed that the supporting metal substrate (Ag(111)) modifies the electronic structure and impacts water diffusivity. On the other hand, the diffusion barrier was substantially higher, around 150 meV, illustrating that the energetics are clearly different compared to the supported 2D-material systems considered here.

We note that our computational methodology differs from the work of Tocci et al. in the fact that we analysed the motion and friction of single water monomers instead of the average motion of a confined water sheet. Thus, higher $\lambda_{GK}$ values are obtained in our study, which can be attributed to the interaction area between the liquid and the graphene/h-BN sheet, where we define $A$ as the area of a water molecule. By assuming an $A$ value corresponding to the full coverage of water molecules on the freestanding material, the $\lambda_{GK}$ values are of the same order of magnitude as in previous works[32,33] ($10.5 \cdot 10^4$ Nsm$^{-3}$ and $3.2 \cdot 10^4$ Nsm$^{-3}$ for h-BN and graphene, respectively). Therefore, the frictional changes experienced upon inclusion of the Ni-support are indeed caused by changes of the surface properties and not any coverage-induced effects.

As shown in Fig. 5c, the different probability distributions (and therefore different corrugation of the PES) for h-BN/Ni and graphene/Ni give rise to completely different motions, which in turn are correlated with the different friction coefficients on the studied 2D systems (Fig. 6d). Phononic effects also contribute to friction, and we thus considered the vibrational density of states (DOS) of the 2D material systems. We calculated the difference between the phonon DOS of the freestanding and Ni-supported systems for both graphene and h-BN, as shown in Fig. 6b, c, respectively. The vibrational power spectrum was calculated from the Fourier transform of the velocity autocorrelation function of the water molecule, with further details described in Supplementary Note 2.7.

The vibrational spectra in Fig. 6b-c show that the coupling between the 2D overlayers (graphene and h-BN) and water changes when the 2D material interacts with the Ni substrate, leading to differences in the distribution of vibrational states between the stretching modes (at $\approx 3800$ cm$^{-1}$) and bending modes (at $\approx 1900$ cm$^{-1}$) - see Supplementary Note 2.7 for the assignment and accuracy of the vibrational states[65,66]. For h-BN, the frequencies of the vibrational bands do not change considerably, while for graphene, the frequency of the 3000 cm$^{-1}$ band decreases upon inclusion of the Ni substrate. Furthermore, for h-BN, as shown in Fig. 6b, only a small dampening of the bending mode at $\approx 1400$ cm$^{-1}$ is observed, while for graphene in Fig. 6c, the bending mode is significantly damped, with a strong redistribution to the stretch mode at 3900 cm$^{-1}$, upon inclusion of the Ni substrate. Comparison of the corresponding peak areas shows that the stretch peak on graphene/Ni is 15 times greater compared to free-standing graphene. Hence, in the case of water on graphene/Ni(111), the substantial increase in the population of the H-O stretching band indicates a stronger molecule-surface coupling with a broader range of available states, with a greater $\lambda_{GK}(t)$ experienced by the molecule. Further analysis of the vibrational power spectrum can be found in Supplementary Note 2.7. As shown in Supplementary Fig. 16, upon inclusion of the Ni(111) substrate, both h-BN and graphene exhibit similar electronic DOS at and around the Fermi level to those without the substrate; thus, we anticipate that the change in friction is not caused by changes in the electronic structure of the material. However, there is a strong correlation between the corrugation of the PES and the corresponding friction, as described above.

Finally, to establish the relationship between the properties of the different 2D systems and $\lambda_{GK}$, we conducted a regression analysis based on properties derived from the time-evolved spatial coordinates of the AIMD trajectories, with statistical significance of the differences being assessed using one-way ANOVA[67], where $p < 0.01$ is considered significant. The strong correlation between the corrugation of the PES and the corresponding friction is demonstrated in Fig. 6d: $\lambda_{GK}$ follows a linear increase as a function of the mean free energy variation $\Delta G$, which describes the overall energy landscape experienced by the water molecule on each of the surfaces. I.e., water molecules situated above the graphene/Ni system encounter a much larger corrugation of the underlying PES according to the mean $\Delta G$, resulting in much larger

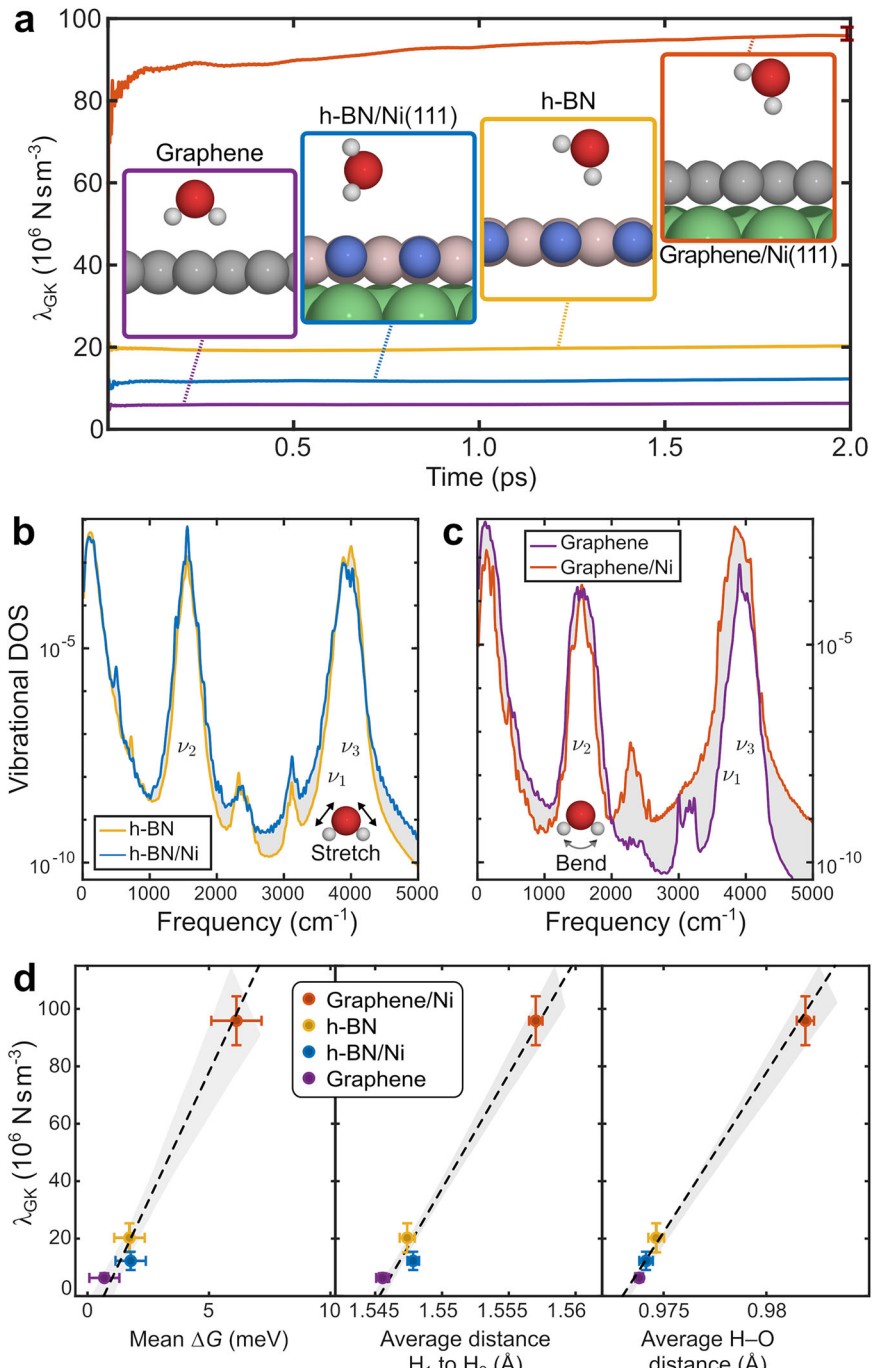

**Fig. 6 | Water friction on 2D material surfaces. a** Friction coefficients estimated from the Green-Kubo relationship (see text) for a single water molecule on graphene and h-BN (see the illustrated insets), showing a distinct change of water friction when Ni is included as a substrate, with the coefficients for graphene/Ni being ≈ 8 times greater than those for h-BN/Ni. This result is in contrast to that for the freestanding substrates, in which graphene exhibits a smaller friction coefficient than h-BN. **b**, **c** show the change in the phonon DOS upon inclusion of the Ni substrate under the same conditions. While the bending mode is slightly damped for both systems with the substrate, for graphene in c, there is a stronger energy redistribution to the stretching mode when the Ni substrate is present (see text for more details). The change in the phonon DOS reflects changes in the water diffusion process and is thus related to the anomalously high friction on graphene/Ni. For all systems, the curves represent an average of the friction coefficients and phonon DOS estimated from AIMD simulations up to 2 ps. Panel **d** illustrates that the increasing friction of the systems is correlated with both the surface topography (the corrugation of the PES) and substrate-induced vibrational changes experienced by the molecules. The linear regression plots, with error bars representing the margin of error (corresponding to the 98% confidence interval to the mean) from 10 independent NVE simulations per system (see Theoretical methods), depict the mean free-energy ($\Delta G$), average H-H distance, and average H-O distance for freestanding h-BN and graphene, as well as for h-BN/Ni and graphene/Ni. Source data are provided as a Source data file.

energy barriers and friction during surface traversal. Based on previous observations of water-surface interactions, which have established a clear correlation between $\Delta G$ and friction[32,34], we illustrate that the supporting substrate plays a crucial role in influencing the motion

of water across the surface of nanomaterials, a facet largely overlooked in prior studies.

Moreover, the distance between the two hydrogen atoms in $H_2O$ is correlated with $\lambda_{GK}$, as shown in Fig. 6d ($R^2 = 0.98$), with an increase in

the mean H-H distance resulting in an increase in $\lambda_{GK}$. This correlation is linked to the vibrational bending modes of the water molecule. Furthermore, $\lambda_{GK}$ is correlated with the O-H bond length ($R^2 = 0.99$), as shown in Fig. 6d, i.e., an increase in the O-H bond length corresponds to an increase in $\lambda_{GK}$ (see also Supplementary Fig. 17). The mean O-H bond length of graphene/Ni is significantly greater than that of freestanding graphene, whereas the mean O-H bond lengths for freestanding h-BN and h-BN/Ni are relatively similar. The observed increase in the bond length is associated with increased energy vibrations due to the O-H bond, accompanied by more extensive stretching vibrations. As both the stretching and bending motions of the water molecule increase, the translational motion is consequently slowed down, resulting in a higher $\lambda_{GK}$.

The atomic-scale friction $\lambda_{GK}$ is thus clearly dependent on both the energy landscape of the surface and the vibrations of the water molecule. Most importantly, considering a dynamic 2D material together with its supporting metal substrate is essential, and our results demonstrate the crucial, yet largely unexplored, role of the supporting substrate in studies of 2D material systems.

## Discussion

In conclusion, we present the first experimental measurements of single-molecule water motion on h-BN/Ni(111). Combined with DFT calculations and AIMD simulations, our results reveal the crucial role of the supporting metal substrate for water diffusion and atomic-scale friction on 2D materials. Comparing water diffusion on h-BN/Ni and graphene/Ni reveals a stark contrast, despite the similar in-plane structures, with a much smaller molecular friction and activation energy for the onset of dynamics on h-BN. Unlike graphene, where water undergoes discrete jumps, water on h-BN follows a quasi-continuous motion, with the molecule walking over the surface and sampling multiple energy minima; thus, translational motion is slowed down, while the barrier for dynamic motion is reduced. By explicitly including the supporting metal substrate in our calculations, we demonstrated that the frictional behaviour of water on h-BN and graphene is reversed in comparison to the freestanding 2D materials, thereby highlighting the essential role of the supporting substrate.

The lower friction of water on h-BN/Ni compared to that on graphene/Ni is an effect of both a decrease in the corrugation of the potential energy surface (PES) and changes in the vibrational coupling. For the latter, we showed that the vibrational coupling between water and h-BN involves a larger contribution from the molecular bending mode than the stretching modes, as in the case of graphene. Our results further suggest that for single-molecule diffusion, friction strongly depends on the internal energy of the molecule, and future studies could investigate a "state-selected" friction, to use terminology employed in reaction dynamics[68–72], to describe the complete dynamics. Understanding the interplay between the PES and vibrational coupling effects on the friction is also essential for designing surfaces with tailored frictional properties and developing advanced nanotechnology materials. E.g., by adjusting the PES corrugation or vibrational coupling, friction on 2D materials may be tuned to enhance lubricity, prevent ice formation, and increase surface hydrophobicity.

To further investigate the influence of the supporting substrate, future work should include single-molecule water studies on the same 2D materials but with different substrates, e.g., by employing graphene grown on SiC(0001)[73]. Inclusion of nonadiabatic effects in the theoretical description, which can account for electron-hole pair excitations[74–76], may further influence the resulting dynamics but have, to our knowledge, been considered only for simple metal surfaces. Future studies may also benefit from an explicit quantification of entropic effects on water diffusion, using thermodynamic integration or harmonic analysis, to complement the enthalpic perspective and fully characterise the free energy landscape[56,77]. Finally, to determine if the observed differences between molecular water on simple metal surfaces

and 2D materials are a general characteristic of the latter, future studies should explore other 2D material systems with varying electronic properties. Notably, while the importance of water-surface interactions has been demonstrated for 2D transition metal dichalcogenides like $WS_2$ and $MoS_2$[78–80], a single-molecule perspective remains lacking.

## Methods

### Experiment and sample preparation

We performed all of our experiments on the Cambridge helium-3 spin echo (HeSE) apparatus, which generates a nearly monochromatic polarised ³He beam with an incident energy of 8 meV, that is scattered off the sample in a fixed 44. 4˚ source-sample-detector geometry. Essentially, the HeSE method is based on the manipulation of the nuclear spin of ³He-atoms in a magnetic field. A detailed description of the setup can be found in refs. 61,81 The schematic principle of HeSE is shown in Fig. 1a. After passing through a magnetic field, the incident helium beam, which is polarised in the $x$ − direction, is split into two wave packets of opposite nuclear spins, which are temporally separated by the spin-echo time $t_{SE}$. The scattered wave packets are recombined in a second magnetic field. As a result of the surface motion that occurs during $t_{SE}$, the two spin components will differ, resulting in a loss of polarisation of the detected beam. An individual HeSE measurement provides the intermediate scattering function $I(\Delta \mathbf{K}, t)$, with $\Delta \mathbf{K}$ the momentum transfer vector. In practice, the analysis is performed for the projection of $\Delta \mathbf{K}$ along a chosen azimuthal direction, such that the ISF is expressed as a function of the scalar $|\Delta K|$. In the case of molecular surface diffusion, as in our study, the ISF follows an exponential decay (see refs. 26,61,82 for more information).

The Ni single crystal has been cleaned by multiple cycles of $Ar^+$ sputtering and annealing to 1050 K. A single layer of h-BN was grown by chemical vapour deposition (CVD) according to the procedure given by Auwärter et al.[51], during which the Ni(111) surface was maintained at 1050 K and exposed to the gas-phase precursor borazine ($B_3H_6N_3$) for a few hours. The hot Ni surface acts as a catalyst, initiating the chemical reactions such as the breaking of the borazine rings, dehydrogenation of borazine, and resulting in the subsequent formation of a complete, non-rotated epitaxial overlayer perfectly matching the Ni(111) lattice due to the small lattice mismatch[47,83]. Additional details can be found in the Sample preparation section in Supplementary Note 1.1.

Water was deposited onto h-BN with a microcapillary array beam doser, which was brought near the surface. To maintain identical experimental conditions throughout each individual measurement, the partial pressure of the water was kept constant by using an automatic leak valve. All dynamic measurements were performed at approximately the same coverage (between 0.12 ML and 0.20 ML, corresponding to an attenuation of the reflectivity by a factor of 4 (see Supplementary Note 1.2). This factor was regularly checked to ensure reproducibility.

### Theoretical methods

Spin-polarised electronic structure calculations were carried out using CASTEP[84], and the Perdew-Burke-Ernzerhof (PBE) functional[85] was used to parameterise the exchange-correlation potential in combination with the Tkatchenko and Scheffler dispersion correction method[86]. All the calculations were performed using Vanderbilt ultrasoft pseudopotentials[87], with a cutoff energy of 400 eV for the plane wave basis set. The Ni(111) substrate was modelled using a 5-layer slab with the lowest two layers fixed to represent both the bulk and surface structures. Calculations were performed on a ($\sqrt{7} \times \sqrt{7}$)R19.1˚ unit cell for the h-BN/Ni(111) and graphene/Ni(111) surfaces and a ($3 \times 3$) unit cell for freestanding h-BN and graphene monolayers to eliminate spurious interactions with periodic images. A vacuum region of 15 Å was introduced to separate the periodically repeated images and avoid spurious interactions. For all calculations, a Monkhorst-Pack[88]($4 \times 4 \times 1$) grid was

used for $k$-point sampling. The self-consistent field energy tolerance was set to $1 \cdot 10^{-7}$ eV. In geometry optimisation using the BFGS minimiser[89], structures were relaxed until the maximum force on each atom was less than 0.025 eV Å$^{-1}$. Adsorption energies ($E_{ads}$) were calculated using the standard formula:

$$E_{ads} = E_{xy} - E_x - E_y \quad , \tag{5}$$

where $E_{xy}$ is the energy of the adsorbed species, $E_x$ and $E_y$ are the energy of the dissociated species. Transition states were identified using the linear-quadratic-synchronous transit (LST/QST) algorithm[90].

The energy barriers reported in Fig. 3, differing by a few meV from the experimental results, lie at the threshold of precision typically attainable with density functional theory (DFT) using generalised gradient approximation (GGA) functionals, such as the Perdew-Burke-Ernzerhof (PBE) functional employed herein. Such minimal energy differences are susceptible to variations arising from numerical convergence criteria, basis set completeness, and the treatment of dispersion interactions, potentially introducing uncertainties on the order of 10–20 meV. Nevertheless, relative energy differences rather than absolute energies, as those between distinct adsorption configurations or diffusion pathways-can often be computed with meV-level accuracy within a consistent DFT framework, provided that stringent convergence parameters are enforced. The close agreement between our calculated energy barriers and the experimental diffusion dynamics observed via HeSE spectroscopy indicates that our DFT approach, augmented with Tkatchenko-Scheffler dispersion corrections, reliably captures the essential $\pi$-H bonding and van der Waals interactions governing water physisorption on h-BN/Ni(111) and graphene/Ni(111). While such quantitative agreement is not routinely expected from GGA-based DFT calculations, analogous accuracies have been reported for related systems dominated by weak intermolecular interactions[44,54,58,91–95]. For the series of AIMD simulations, used to investigate the dynamic behaviour of water molecules at the surface, the velocity Verlet algorithm[96], as implemented by CASTEP[84] was employed. The system was initialised with the coordinates set to the global minimum position, and initial velocities were randomised using single iteration AIMD simulations within the canonical ensemble (NVT). The AIMD trajectories were extended for a minimum duration of 2 ps (to ensure $\lambda_{GK}(t)$ had converged) with a timestep of 1 fs, and the temperature converged at ≈150 K within the microcanonical ensemble (NVE). To ensure comprehensive analysis, AIMD simulations that extended beyond 2 ps were segmented into 2 ps intervals. A minimum of 10 AIMD trajectories were generated for each system. ($\sqrt{7} \times \sqrt{7}$)R19.1° unit cells of h-BN/Ni(111) and graphene/Ni(111) were considered, as well as the ($3 \times 3$) expanded, free-standing layers of graphene and h-BN. In each timestep of the AIMD trajectories, the centre of mass of the water molecule was determined, and the closest surface site was calculated, accounting for the dynamic movement of the underlying surface atoms. By aggregating data from all AIMD simulations conducted for a specific system, we are able to derive the probability of the water being located at distinct surface sites. For comparison, to evaluate the probability distribution produced using all relaxed geometries of h-BN/Ni(111) during the construction of the PES (Fig. 2), the representing configuration space was sampled at different temperatures according to Boltzmann population analysis (Supplementary Fig. 10). The Supplementary Movies show the full length of the AIMD simulation, with every frame representing a 1 fs timestep, played at 120 frames per second.

## Data availability
The data that support the findings of this study are available from the TU Graz repository[97] and from the corresponding authors upon request. Source data are provided with this paper.

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

## Acknowledgements

This research was funded in whole, or in part, by the Austrian Science Fund (FWF) [10.55776/P34704]. M.S. acknowledges the Royal Society for funding his research through the University Research Fellowship URF/R/191029 and the UK's HEC Materials Chemistry Consortium, which is funded by EPSRC (EP/R029431), for time on the ARCHER UK National Supercomputing Service. L.S. acknowledges funding from the John Templeton Foundation grant number 62210. The authors acknowledge the use of and support by the Cambridge Atom Scattering Facility (https://atomscattering.phy.cam.ac.uk) and EPSRC award EP/T00634X/1.

## Author contributions

P.S. and A.T. conceived the experiments. P.S. conducted the measurements, and P.S. and A.T. analysed the data. A.P. performed the DFT calculations, A.P. and N.X. conducted the ab initio molecular dynamics simulations, calculation of friction coefficients, and N.X. electronic DOS calculations, and L.S. conducted the vibrational DOS. A.P., N.X., and L.S. analysed the ab initio results. M.S. and A.T. supervised the project and were responsible for funding acquisition. All authors discussed the results and wrote the manuscript.

## Competing interests

The authors declare no competing interests.
