## [Transparent Peer Review file · Nature Communications]

Understanding Water Behaviour on 2D Material Interfaces through Single-Molecule Motion on h-BN and Graphene

Corresponding Author: Dr Marco Sacchi

Version 0:

Reviewer comments:

Reviewer #1

(Remarks to the Author)

In „Single-Molecule Water Motion on h-BN and Graphene: A paradigm shift in Understanding the Behaviour of Water on 2D Material Interfaces“, P. Seiler et al. use He-spin echo measurements and ab initio calculations to compare the diffusivity of monomeric water on four related surfaces, Ni supported and non-supported graphene and boron nitride and its influence on friction. The article is well-written, and the data is sound and thoroughly analyzed. The advantage of He-spin echo versus another common method of investigating diffusivity, scanning tunnelling microscopy, is the much higher experimental temperature and the statistical data obtained in this involved and challenging technique. The topic is timely, and the results deserve publication in Nature Communications. However, the claim that "such a behaviour of a single water molecule had not been reported previously" is incorrect. A conceptionally similar adsorption, including hydrogen bonding in addition to surface interaction with the oxygen atom of water, and a motion, including a combination of rotation and translation, has been reported for water adsorption and diffusion on NaCl on Ag(111) (P. Cabrera-Sanfelix and co-authors, J. Phys. Chem. B 2006, 110, 24559, ACS Nano 9, 3572–3578 (2015)). Also, the influence of the metal support on the diffusivity of a bilayer of NaCl was discussed and compared to bulk NaCl. BN and graphene exist free-standing, in contrast to NaCl, and the previous studies did not relate the single molecule diffusivity to friction. Nonetheless, the previous studies need to be cited, discussed, and related to the present study. In this respect, the title claiming a 'paradigm shift' seems exaggerated. Even without this claim, the study remains relevant because He-Spin echo probes a largely different temperature than STM and because of the relevance to 'true' 2D material and relation to friction.

Further major points to address:

The desorption energy was determined based on the Redhead formula to be approximately (0.53 ± 0.02) eV. First, the Redhead formula is so approximate that an error bar of 20 meV seems inadequate. More importantly, the formula holds for 1st order desorption, for which the desorption peak maximum is independent of initial coverage. Water desorbs from a large number of weakly interacting surfaces with zero-order desorption. The authors present their desorption only for one coverage. Please include a desorption experiment for another coverage in the supplementary material to confirm 1st order desorption and, thus, that the method is applicable.

Island formation does not exist up to 125 K. This is surprising given the very low diffusion barrier of water monomers on BN, suggesting a classical nucleation-growth scenario according to Venables theory. Is there an additional barrier for water dimer formation or attachment of water to existing clusters? Please comment on this apparent discrepancy.

P.M. is named in the Author contributions, who conducted the experiments. P.M. is not on the author list. P.M. should be a co-author unless it is a typo.

Minor points to consider:

A multi-step motion on BN should alter the Arrhenius prefactor from the single-step motion on graphene. Please comment on it if the data exists. If not, this might be beyond the scope of the present article but is interesting from a fundamental point of view of diffusivity.

The AIMD simulation presented in Fig. 5 suggests a directed motion, for which the molecule does not fully equilibrate in its new energetic minimum between single diffusion steps and not a diffusive, i.e., random, motion. It might be an artefact of the used simulation, but should be commented on if all trajectories are directed as the one shown. Or, diffusive trajectories could be added to the supplementary information. If these 'long' jumps are part of the motion, how is this accounted for in the He-spin-echo analysis? Please clarify this point for the less specialized reader.

Inconsistencies in the Journal naming in the reference list will be corrected for the main text by the Editors. The authors should unify journal abbreviations and capitalization in the supplementary material. Also, ensure a space between the

number and the unit in the text of the supplementary material.

Reviewer #2

(Remarks to the Author)

The manuscript "Single-Molecule Water Motion on h-BN and Graphene: A Paradigm Shift in Understanding the Behaviour of Water on 2D Material Interfaces" by P. Seiler et al. reports about the diffusion of water molecules at sub-monolayer regime on h-BN and graphene. The experimental study is carried out by using the helium-3 spin echo technique and the results show a difference in the behavior of water on the two substrates. For the interpretation of data, the authors performed DFT calculations and ab-initio molecular dynamics simulations for estimating adsorption and activation energies. Their conclusion is: the model in which the hopping of point-like particles between two different lattice sites can reproduce the data on graphene but it is not able to reproduce the data on h-BN for the entire range of momentum exchange probed in the experiment. In particular the authors observe that the trend of experimental points at small momentum exchange is related to a dynamic process involving molecular translation and rotation.

In my opinion, the paper is technically sound, the manuscript clearly written with proper details, there are new results, but I have some points that the authors should address.

1) The authors consider a single sigma to estimate the error bars throughout the manuscript and I think this value is optimistic. Moreover, in figure 3a, the activation energy is estimated in a limited range of temperature (120-135 K) for 3 points along ΓM and two points along ΓK . In SI Figure 4, authors show thermal desorption measurements, water film desorbs around 170 K and seems stable up to 150 K, therefore an extension up to 150 K of the Arrhenius plot looks possible and more points in a extended temperature range would increase the reliability of the estimation even with larger error bars.

2) In the same figure 3, there is a comparison among calculated energy barriers which differ of a few meV, a value too small to be calculated with the necessary precision (and accuracy) by DFT.

3) In figure 4, the dephasing rate alpha is plotted for h-BN and graphene. The authors comment about the different trend at small momentum exchange for the two cases ($DK < 1 \text{ \AA}^{-1}$). A careful inspection of the two plots reveals that the h-BN case presents almost a constant value with points between 3 and 5 ns⁻¹, on the other hand, even in the graphene case the points are not tending to zero for DK going to zero but to a non-vanishing constant value. At this point, based on the experiment, even for graphene a small contribution of the molecular rotation cannot be excluded.

4) In SI Figure 1 the specular reflection intensity shows a decay which is monotonic and exponential for 120 K, but for 130 K there is a clear initial (apparent) linear decay, followed by an exponential decay. The initial part is related to the reduction of reflectivity since clean surface presents higher reflectivity with respect to regions covered by the adsorbate (producing diffuse scattering) and, during the deposition, the percentage of clean surface is reduced causing a reduction of specular intensity. The adsorbate islands grow until they start to touch and merge and the intensity trend could change. In the present case, the "linear" decay extends up to 0.03 ML, a value which is very small compared to other cases present in literature (up to 20% of a ML), moreover, a merging of islands at this low 0.03 ML coverage is not justifiable, even taking into account the effective cross section seen by helium. Therefore some doubts arise about the calibration procedure to estimate the coverage. Maybe the measurement of the pressure in a specific position of the chamber is not real pressure at the sample. The authors should try to estimate a reasonable error bar (or confidence interval) on coverage and, maybe, increase the 0.16 ML value.

5) In figure 6a, friction coefficient $\lambda(t)$ are reported, my concern is about the graphene/Ni(111) case that look like its value has not yet reached its asymptotic value at 2 ps.

Minor correction:

Main text:

- 1) in "Walking motion of Water on h-BN": " in canonical simulations ((NVT)" remove double parenthesis
- 2) In "Theoretical methods" the unit cell notation " $(\sqrt{7} \times \sqrt{7})R19.1^\circ$ " has to be corrected to " $(\sqrt{7} \times \sqrt{7})R19.1$ "
- 3) the meaning of " periodic images were eliminated. " is not very clear.

In SI

- 1) 1.2 Coverage calibration, line 39 "In Supplementary Supplementary Figure 1", remove repetition
- 2) page 2, line 76: "v is the preexponential or frequency factor," v is missing in the equation
- 3) page 4, line 165 & line 178 "hBN"-> "h-BN"

In conclusion, the manuscript require changes before to be accepted for publication.

Reviewer #3

(Remarks to the Author)

The authors combine helium spin-echo spectroscopy and ab initio calculations to show that single water molecules on h-BN/Ni(111) diffuse via a coupled rotational-translational "walking" mechanism—with an activation energy roughly 2.5× lower than the ~60 meV barrier on graphene/Ni—despite exhibiting similar binding energies on both substrates. They

further demonstrate that metal support dramatically alters friction, with h-BN/Ni displaying markedly lower water friction than graphene/Ni, challenging classical jump-between-site models and underscoring the critical role of substrate polarity in designing 2D-material-based microfluidic systems.

The work is very interesting, it addresses the lack of data for the H₂O on h-BN/Ni system, and it shows marked differences between h-BN and its graphene counterpart case. It is shown that despite the similarity of the substrate and H₂O adsorption energy on h-BN/Ni and graphene/Ni, both experiments and theory show completely different motions of single water molecules on the two surfaces. However, there are a couple comments that need to be addressed before publication.

1. What would be the role of entropy in the energetics described in this work? Is entropy accounted for?
2. DFT methods used here are known to overestimate adsorption energies, especially compared to coupled-cluster/QMC methods. How would this overestimation affect calculated diffusion barriers?
3. The calculated vibrational spectra for the OH stretching is in the order of 3900 cm⁻¹. This is much higher than normally found for experimental OH stretches (typically below 3750 cm⁻¹). Why is this calculated frequency much higher than experimentally found?
4. In the following sentence in page 2, "A previously reported HeSE study of water motion on graphene illustrated the stark contrast between water diffusion on 2D materials such as graphene and metal substrates, where water is bound much stronger and further influenced by hydrogen bonding between the molecules." It is not clear in the sentence to which case it refers that water is bound much stronger. Does this refer to the latter (metal surface)?

Reviewer #4

(Remarks to the Author)

This is a most interesting paper. It combines state-of-the-art experimentation and simulation to probe the nature of the surface water interactions on h-BN and graphene. The present referee is not an expert in the experimental techniques but they appear to have carefully undertaken. The modelling work is of high quality using state of the art techniques and is clearly reported and reproducible. The results are particularly interesting, especially in the contrast between the diffusion mechanisms on the two systems. The proposed mechanism on h-BN is of special interest although I think it could be more clearly explained. The paper is well written and will be of general interest. Publication in Nat Comms is recommended but I suggest two modifications:

1. Can we have a clearer account of the rotational- translational dynamic motion of water, with perhaps a good figure.
2. Can we remove "paradigm shift" from the title. This is very interesting and novel work but it is not a paradigm shift in understanding water at 2D interfaces

Version 1:

Reviewer comments:

Reviewer #1

(Remarks to the Author)

The author has thoughtfully addressed all my comments, responding to them in a satisfactory manner and revising the manuscript appropriately. I fully endorse the acceptance of this article for publication in Nature Communications in its current form. I sincerely considered all my remarks. They have answered them to my full satisfaction and altered the manuscript accordingly.

I fully support the acceptance of this article for publication in Nature Communications in the present form.

Reviewer #2

(Remarks to the Author)

The revised version of the article with changed title "Single-Molecule Water Motion on h-BN and Graphene: Understanding the Behaviour of Water on 2D Material Interfaces" now addresses all the points I raised in my previous review and I am satisfied with the authors' changes. In my opinion, this revised version can be accepted for publication.

Reviewer #4

(Remarks to the Author)

The authors have responded positively and well to my comments and those of the other referees. The additional information, especially on mechanistic aspects is helpful and the new title is more appropriate.

I recommend that the paper be accepted for publication

Reviewer #1 (Remarks to the Author):

In „Single-Molecule Water Motion on h-BN and Graphene: A paradigm shift in Understanding the Behaviour of Water on 2D Material Interfaces“, P. Seiler et al. use He-spin echo measurements and ab initio calculations to compare the diffusivity of monomeric water on four related surfaces, Ni supported and non-supported graphene and boron nitride and its influence on friction. The article is well-written, and the data is sound and thoroughly analyzed. The advantage of He-spin echo versus another common method of investigating diffusivity, scanning tunnelling microscopy, is the much higher experimental temperature and the statistical data obtained in this involved and challenging technique. The topic is timely, and the results deserve publication in Nature Communications.

Author reply:

We thank the referee for their constructive suggestions and for recognising the timeliness and significance of our work. We are pleased that the results were found to be suitable for publication in Nature Communications, and we have addressed all comments in detail below.

Major Points

1) “However, the claim that “such a behaviour of a single water molecule had not been reported previously” is incorrect. A conceptionally similar adsorption, including hydrogen bonding in addition to surface interaction with the oxygen atom of water, and a motion, including a combination of rotation and translation, has been reported for water adsorption and diffusion on NaCl on Ag(111) (P. Cabrera-Sanfeliix and co-authors, *J. Phys. Chem. B* 2006, 110, 24559, *ACS Nano* 9, 3572–3578 (2015)). Also, the influence of the metal support on the diffusivity of a bilayer of NaCl was discussed and compared to bulk NaCl. BN and graphene exist free-standing, in contrast to NaCl, and the previous studies did not relate the single molecule diffusivity to friction. Nonetheless, the previous studies need to be cited, discussed, and related to the present study. In this respect, the title claiming a ‘paradigm shift’ seems exaggerated. Even without this claim, the study remains relevant because He-Spin echo probes a largely different temperature than STM and because of the relevance to ‘true’ 2D material and relation to friction.”

Author reply:

We thank the reviewer for the constructive comment and for drawing attention to the relevant prior work on single water molecule dynamics on NaCl surfaces. In response, we have cited these studies as Ref. [25] and [61] in the revised manuscript and have incorporated a discussion of their relevance, as detailed below.

Cabrera-Sanfeliix *et al.* (*J. Phys. Chem. B* 2006, 110, 24559) examined the adsorption and diffusion of a single water molecule on NaCl(001) using density functional theory (DFT). Their results revealed a concerted translation–rotation diffusion mechanism, facilitated by electrostatic interactions with Na⁺ and hydrogen bonding with Cl⁻. While the quasi-continuous motion they describe is conceptually analogous to our findings, the underlying substrate is ionic rather than covalent or weakly polar, as in the h-BN and graphene surfaces studied here. Moreover, this study does not address the role of friction or metal support effects, which are central to our investigation.

In *ACS Nano* 2015, 9, 3572–3578, Cabrera-Sanfeliix and co-authors employed scanning tunnelling microscopy (STM) and DFT to study water dynamics on a NaCl bilayer supported on Ag(111). This work demonstrates that the metal substrate modifies adsorption energies and diffusion pathways, an effect that aligns qualitatively with our finding of Ni(111)-induced

modulation of potential energy surface corrugation. However, the focus remains on STM-accessible dynamics at lower temperatures (42–52 K), whereas our work probes single-molecule dynamics at elevated temperatures (120–140 K) via helium spin-echo (HeSE) spectroscopy.

Our study extends beyond these prior contributions in several critical aspects:

1. We employ HeSE to investigate single-molecule water dynamics on h-BN and graphene above 100 K, thus accessing a higher temperature regime and a different time scale (picoseconds) than STM.
2. We quantify atomic-scale friction via the Green-Kubo formalism, a method not utilised in the NaCl studies.
3. We demonstrate a substrate-induced inversion of frictional response between h-BN and graphene when supported on Ni(111), revealing a previously unreported metal–2D material coupling effect.
4. Our system is based on covalently bonded 2D materials rather than ionic substrates, introducing fundamentally different interfacial interactions and electronic environments.

To reflect these distinctions, we have revised the title and updated the introduction and discussion sections to acknowledge and appropriately contextualise the prior NaCl studies.

Title Revision

Original Title: “Single-Molecule Water Motion on h-BN and Graphene: A Paradigm Shift in Understanding the Behaviour of Water on 2D Material Interfaces”

Revised Title: “Single-Molecule Water Motion on h-BN and Graphene: Understanding the Behaviour of Water on 2D Material Interfaces”

Revised text

Page 1 (Introduction):

However, while properties such as wettability and ice nucleation at the macroscopic level are well understood,²⁰ measurements at the atomic or molecular level are scarce and focused on metal substrates^{21–24} or metal-supported thin ionic films such as NaCl²⁵.

Page 5 (Motion):

The observation of such a quasi-continuous, “walking” motion of water on h-BN/Ni(111), characterised by a combination of translational and rotational dynamics, shares conceptual similarities with prior studies of single-molecule water diffusion on NaCl(001) in Refs^{25,61}: Water on a NaCl bilayer on Ag(111) undergoes a rocking motion, where translations are coupled with rotation, driven by electrostatic interactions with Na⁺ ions and hydrogen bonding with Cl[−] ions.²⁵

Page 7 (Friction and role of the substrate):

Interestingly, in a previous study of water on NaCl bilayers, the role of the underlying substrate was also highlighted - specifically, the Ag(111) substrate was found to significantly affect water adsorption on NaCl/Ag(111) compared to bulk NaCl.²⁵ Although the study did not address friction, it clearly showed that the supporting metal substrate (Ag(111)) modifies the electronic structure and impacts water diffusivity. On the other hand, the diffusion barrier was substantially higher, around 150 meV, illustrating that the energetics are clearly different compared to the supported 2D-material systems considered here.

Further major points to address:

The desorption energy was determined based on the Redhead formula to be approximately (0.53 ± 0.02) eV. First, the Redhead formula is so approximate that an error bar of 20 meV seems inadequate. More importantly, the formula holds for 1st order desorption, for which the desorption peak maximum is independent of initial coverage. Water desorbs from a large number of weakly interacting surfaces with zero-order desorption. The authors present their desorption only for one coverage. Please include a desorption experiment for another coverage in the supplementary material to confirm 1st order desorption and, thus, that the method is applicable.

Author reply:

We appreciate the referee's insightful comment concerning the desorption kinetics and the limitations of the Redhead analysis. We agree that the Redhead method assumes first-order desorption and that its precision is inherently limited, particularly in the context of water desorption on weakly interacting surfaces such as hexagonal boron nitride (h-BN). The main goal of including this analysis was to provide a reference point for the desorption temperature regime, as this represents, to our knowledge, the first such experimental data reported for water on h-BN.

Although the desorption kinetics are not the central focus of our study, and our main conclusions regarding diffusion are unaffected by this discussion, we acknowledge the importance of clarifying the desorption order. Unfortunately, we do not have additional thermal desorption spectra for different coverages. Nevertheless, we have experimentally confirmed the desorption temperature regime through the recovery of the helium reflectivity signal.

To address the referee's concern, we have substantially revised the supplementary section. Specifically, we now discuss the potential for zero-order desorption kinetics, which are more commonly observed for water on weakly interacting surfaces such as graphene, due to the formation of hydrogen-bonded water clusters – and calculate also the energy based on that kinetics. We also increase the uncertainty in the derived desorption energy to reflect the approximations inherent in both the first- and zero-order analyses.

The reported value in the main text has been corrected; for the added paragraphs see the revised version of the supplementary information – on page 2.

Island formation does not exist up to 125 K. This is surprising given the very low diffusion barrier of water monomers on BN, suggesting a classical nucleation-growth scenario according to Venables theory. Is there an additional barrier for water dimer formation or attachment of water to existing clusters? Please comment on this apparent discrepancy.

Author reply:

We clarify that our manuscript does not claim an absence of island formation below 125 K and we have also done additional calculations for dimer formation and diffusion, see below.

Added on page 2:

while increasing the temperature to 135 K prevents full surface coverage, indicating hydrophobic behaviour and island formation on h-BN/Ni, similar to water adsorption on graphene/Ni.⁴³ As such it does not exclude the formation of clusters or islands; however, once small aggregates such as dimers have formed, they are unlikely to dissociate, and, as discussed below, exhibit significantly reduced diffusivity. The elastic component, from the

In response, we have now expanded the discussion of dimer diffusion and dissociation in a newly added Section 2.2 of the Supplementary Information. As shown in Supplementary Figures 7 and 8, water dimers on both freestanding and Ni-supported h-BN exhibit significantly higher energy barriers for diffusion (~80–100 meV) and dissociation (220 meV) compared to isolated water molecules. These elevated barriers indicate that dimers are effectively immobile under the experimental conditions and thus do not contribute to the observed dynamics. The comparison confirms that the experimentally measured activation energies are best represented by single-molecule behaviour.

We have further added the following text on page 4 of the main manuscript to point the reader to the SI:

For completeness, we also examined the possibility of dimer formation, dissociation, and diffusion, as detailed in Section 2.2 of the Supplementary Information and illustrated in Supplementary Figures 7 and 8. While dimer formation is thermodynamically allowed under low-coverage conditions, the associated diffusion barriers are substantially higher than those for single-molecule motion. Specifically, calculated barriers for dimer translation and rotation range from 79 to 220 meV, in contrast to the 24 meV barrier observed experimentally for monomer diffusion. Moreover, dimer dissociation involves an energy cost of approximately 220 meV, indicating that dimers, once formed, are kinetically stable but effectively immobile under the experimental conditions. These findings support the conclusion that the observed dynamics predominantly arise from isolated water molecules and that single-molecule simulations provide the most accurate description of the experimental system.

P.M. is named in the Author contributions, who conducted the experiments. P.M. is not on the author list. P.M. should be a co-author unless it is a typo.

Author reply:

We would like to apologise for this typo. P. M. is Philipp Maier which at the time of writing was his birth surname - after getting married this has now changed to Philipp Seiler. We have corrected this accordingly in the author's contribution.

Minor points:

1) "A multi-step motion on BN should alter the Arrhenius prefactor from the single-step motion on graphene. Please comment on it if the data exists. If not, this might be beyond the scope of the present article but is interesting from a fundamental point of view of diffusivity.

Author reply:

We appreciate the reviewer's observation concerning the possible impact of multi-step motion on the Arrhenius prefactor. The experimentally determined dephasing rate $\alpha(\Delta K)$ in helium spin-echo measurements is a reciprocal-space observable that integrates over all dynamical processes in real space. Consequently, it does not allow for a direct decomposition into translational, rotational, or multi-step components. In this framework, the extracted Arrhenius parameters represent effective quantities that encapsulate the ensemble-averaged dynamics projected onto a single timescale.

We are, of course, mindful that within the framework of transition state theory, a multi-step diffusion mechanism can lead to an enhanced entropy of the transition state, thereby increasing the Arrhenius prefactor. This arises due to the greater number of accessible microstates and vibrational modes contributing to the rate-limiting barrier crossing. Conversely, enhanced frictional coupling to the substrate or internal degrees of freedom may act to suppress the effective attempt frequency. Furthermore, for strongly corrugated potential energy surfaces (PES), the classical picture of single-barrier crossing becomes less applicable, and the prefactor becomes highly sensitive to the detailed nature of the PES and the coupling to dissipative environments.

Given these considerations, we note that a more detailed analysis of prefactor variations arising from multi-step dynamics lies beyond the scope of the present study, whose primary focus is on quantifying the effective activation energy and the overall diffusive behaviour – and we have added the following sentence to the manuscript on page 4:

rotations. We note that such multi-step diffusion pathways can, in principle, alter the Arrhenius prefactor through entropic contributions to the transition state, although such effects are not resolved in the experimental dephasing observable $\alpha(\Delta K)$, and in general both prefactor and activation energy may vary with temperature, as shown e.g. in Ref.⁵² The molecule first crosses a boron site,

2) “The AIMD simulation presented in Fig. 5 suggests a directed motion, for which the molecule does not fully equilibrate in its new energetic minimum between single diffusion steps and not a diffusive, i.e., random, motion. It might be an artefact of the used simulation, but should be commented on if all trajectories are directed as the one shown. Or, diffusive trajectories could be added to the supplementary information. If these 'long' jumps are part of the motion, how is this accounted for in the He-spin-echo analysis? Please clarify this point for the less specialized reader.”

Author reply:

We thank the reviewer for the careful observation. The trajectory presented in Figure 5 is intended as a representative illustration of water mobility on h-BN/Ni(111), rather than as an exemplar of universal behaviour across all simulations. While the specific trajectory shown may appear directed, examining the motion in detail shows it is random, consistent with thermally activated diffusion processes.

To facilitate a clearer understanding of the dynamic behaviour, we have provided visualisations of several ab initio molecular dynamics (AIMD) trajectories in the form of supplementary videos. These include three additional trajectories that accompany the revised manuscript and are referenced explicitly in the updated text. The trajectory shown in Figure 5 is designated as Video 1.

All videos (10 in total) are also permanently archived in the data repository cited under the "Data Availability" section, enabling full access to the underlying motion data.

We further note that the mentioned phenomenon is consistent with finite-temperature ab initio molecular dynamics at 150 K: Thermal energy permits the molecule to explore a range of configurations around local minima without necessarily achieving full equilibration within each potential well prior to a subsequent transition. This behaviour stands in contrast to the situation at approximately 0 K, where atomic positions relax strictly to the nearest local minimum in a deterministic manner, governed solely by the potential energy landscape and not by thermal fluctuations.

This form of motion is expected for thermally activated dynamics where the thermal energy of a system at constant T is comparable to the energy barrier and has been well documented in both theoretical and experimental studies^{1,2}.

3) “Inconsistencies in the Journal naming in the reference list will be corrected for the main text by the Editors. The authors should unify journal abbreviations and capitalization in the supplementary material. Also, ensure a space between the number and the unit in the text of the supplementary material.”

Author reply:

We have checked and updated the supplementary references, and we fixed the missing space between number and text.

¹ <https://doi.org/10.1103/PhysRevLett.96.026101>

² <https://pubs.acs.org/doi/full/10.1021/jacs.5b08001>

Reviewer #2 (Remarks to the Author):

The manuscript "Single-Molecule Water Motion on h-BN and Graphene: A Paradigm Shift in Understanding the Behaviour of Water on 2D Material Interfaces" by P. Seiler et al. reports about the diffusion of water molecules at sub-monolayer regime on h-BN and graphene. The experimental study is carried out by using the helium-3 spin echo technique and the results show a difference in the behavior of water on the two substrates. For the interpretation of data, the authors performed DFT calculations and ab-initio molecular dynamics simulations for estimating adsorption and activation energies. Their conclusion is: the model in which the hopping of point-like particles between two different lattice sites can reproduce the data on graphene but it is not able to reproduce the data on h-BN for the entire range of momentum exchange probed in the experiment. In particular the authors observe that the trend of experimental points at small momentum exchange is related to a dynamic process involving molecular translation and rotation.

In my opinion, the paper is technically sound, the manuscript clearly written with proper details, there are new results, but I have some points that the authors should address.

Author reply:

We thank the referee for the positive assessment of our work and for acknowledging the technical soundness, clarity, and novelty of our results. We appreciate the thoughtful comments and suggestions, which have helped us to improve the manuscript. Below, we address each point raised in detail.

1) The authors consider a single sigma to estimate the error bars throughout the manuscript and I think this value is optimistic. Moreover, in figure 3a, the activation energy is estimated in a limited range of temperature (120-135 K) for 3 points along ΓM and two points along ΓK . In SI Figure 4, authors show thermal desorption measurements, water film desorbs around 170 K and seems stable up to 150 K, therefore an extension up to 150 K of the Arrhenius plot looks possible and more points in a extended temperature range would increase the reliability of the estimation even with larger error bars.

Author reply:

We thank the referee for raising this important point regarding the uncertainty estimates and the limited temperature range used for the Arrhenius analysis. We agree that the determination of the activation energy from a narrow temperature interval can introduce sensitivity to statistical and systematic errors.

While we did conduct an additional measurement at 140 K, we note that this temperature lies near the onset of water desorption, as confirmed by the thermal desorption profile presented in SI Figure 4. Under these conditions, maintaining a constant surface coverage becomes increasingly difficult without applying a continuous water overpressure, which is experimentally challenging and introduces further uncertainties.

The temperature range employed in our analysis (120–135 K) was selected to ensure that diffusion dominates over desorption, thereby maintaining well-defined coverage conditions. It is worth noting that similar or even narrower temperature windows are frequently used in scanning tunnelling microscopy (STM) studies of water diffusion, due to desorption constraints, but also to achieve the necessary temporal resolution.

We have already combined two independent datasets to maximise the temperature coverage and statistical reliability of the extracted activation energy. Given the limitations described above, we believe this range represents the most robust basis for Arrhenius analysis under the current experimental conditions. Nevertheless, to reflect the acknowledged uncertainties

more conservatively, we have increased the reported error bars on the activation energy estimate in the revised manuscript, and we have also added 2 sentences on page 4:

We note that due to the proximity of the diffusion and desorption regimes, the temperature range is constrained to 120-135 K, within which coverage remains stable. To reduce the confidence interval of the linear fit in Fig. 3(a), we include data points for both the $\bar{\Gamma}\bar{M}$ (blue) and $\bar{\Gamma}\bar{K}$ (green) azimuths, assuming that diffusion occurs at a similar rate along both directions, as also supported by AIMD calculations.

2) In the same figure 3, there is a comparison among calculated energy barriers which differ of a few meV, a value too small to be calculated with the necessary precision (and accuracy) by DFT.

Author reply:

We appreciate the reviewer's observation regarding the limitations in absolute energy accuracy inherent to density functional theory (DFT), particularly when using generalised gradient approximation (GGA) functionals such as the Perdew-Burke-Ernzerhof (PBE) functional employed in our study. While absolute quantities - such as adsorption energies - are often subject to systematic errors and uncertainties arising from numerical convergence, basis set limitations, and dispersion treatment, relative energy differences between local minima and transition states within the same DFT framework can be determined with a precision of a few meV, provided that computational parameters are carefully converged. This distinction is particularly relevant when comparing diffusion barriers or competing adsorption configurations within a unified computational approach and the close agreement between our calculated energy barriers and the experimental values obtained from helium spin-echo (HeSE) measurements suggests that our DFT calculations effectively capture the key physical interactions governing water molecule dynamics on h-BN/Ni(111) and graphene/Ni(111).

In this context, the close agreement between our calculated energy barriers and the experimental values obtained via helium spin-echo (HeSE) spectroscopy suggests that our DFT calculations accurately capture the essential physics underlying water molecule dynamics on h-BN/Ni(111) and graphene/Ni(111). The governing interactions-namely π -H bonding and weak van der Waals forces-are effectively described by the GGA functional in conjunction with the Tkatchenko-Scheffler dispersion correction. Furthermore, to substantiate the reliability of our methodology, we have cited relevant benchmark studies that demonstrate similar levels of accuracy for weakly interacting surface systems [87,90]. Accordingly, while absolute agreement within a few meV is not generally expected, the consistency observed in our study reflects the robustness of our approach in capturing the relative energetics of water physisorption and diffusion on these two-dimensional substrates as also shown in Refs [43,53,56,88,89,91].

We have also added the following paragraph in the manuscript:

The energy barriers reported in **Fig. 3**, differing by a few meV from the experimental results, lie at the threshold of precision typically attainable with density functional theory (DFT) using generalised gradient approximation (GGA) functionals, such as the Perdew–Burke–Ernzerhof (PBE) functional employed herein. Such minimal energy differences are susceptible to variations arising from numerical convergence criteria, basis set completeness, and the treatment of dispersion interactions, potentially introducing uncertainties on the order of 10–20, meV. Nevertheless, relative energy differences rather than absolute energies, as those between distinct adsorption configurations or diffusion pathways can often be computed with meV-level accuracy within a consistent DFT framework, provided that stringent convergence parameters are enforced. The close agreement between our calculated energy barriers and the experimental diffusion dynamics observed via helium spin-echo (HeSE) spectroscopy indicates that our DFT approach, augmented with Tkatchenko–Scheffler dispersion corrections, reliably captures the essential π -H bonding and van der Waals interactions governing water physisorption on h-BN/Ni(111) and graphene/Ni(111). While such quantitative agreement is not routinely expected from GGA-based DFT calculations, analogous accuracies have been reported for related systems dominated by weak intermolecular interactions.^{43,53,57,89–93}

3) In figure 4, the dephasing rate α is plotted for h-BN and graphene. The authors comment about the different trend at small momentum exchange for the two cases ($DK < 1 \text{ \AA}^{-1}$). A careful inspection of the two plots reveals that the h-BN case presents almost a constant value with points between 3 and 5 ns⁻¹, on the other hand, even in the graphene case the points are not tending to zero for DK going to zero but to a non-vanishing constant value. At this point, based on the experiment, even for graphene a small contribution of the molecular rotation cannot be excluded.

Author reply:

We thank the reviewer for this careful inspection and insightful observation. Indeed, as the reviewer points out, the dephasing rate α for graphene does not vanish entirely in the low- ΔK limit. While we cannot categorically exclude a minor contribution from rotational degrees of freedom for H₂O adsorbed on graphene, as discussed in Ref [43] and Supplementary material thereof, the constant offset is mostly an artifact of the correction upon extracting the self-diffusion rather than the collective diffusion on graphene where the latter has a strong signature of repulsive interactions. We have also added the following sentence in the manuscript on page 5:

the constant offset of $\alpha(\Delta K)$. For comparison, **Fig. 4(b)** displays $\alpha(\Delta K)$ for graphene/Ni, which exhibits a negligible contribution from rotational motion relative to h-BN, and is instead dominated by repulsive intermolecular interactions, for a detailed analysis, see Refs.^{43,61} and its SI.

4) In SI Figure 1 the specular reflection intensity shows a decay which is monotonic and exponential for 120 K, but for 130 K there is a clear initial (apparent) linear decay, followed by

an exponential decay. The initial part is related to the reduction of reflectivity since clean surface presents higher reflectivity with respect to regions covered by the adsorbate (producing diffuse scattering) and, during the deposition, the percentage of clean surface is reduced causing a reduction of specular intensity. The adsorbate islands grow until they start to touch and merge and the intensity trend could change. In the present case, the "linear" decay extends up to 0.03 ML, a value which is very small compared to other cases present in literature (up to 20% of a ML), moreover, a merging of islands at this low 0.03 ML coverage is not justifiable, even taking into account the effective cross section seen by helium. Therefore some doubts arise about the calibration procedure to estimate the coverage. Maybe the measurement of the pressure in a specific position of the chamber is not real pressure at the sample. The authors should try to estimate a reasonable error bar (or confidence interval) on coverage and, maybe, increase the 0.16 ML value.

Author reply:

We appreciate the reviewer's detailed analysis and agree that pressure- and coverage-calibration procedures are intrinsically challenging. We follow the methodology outlined in Ref [43] and its SI and have added additional clarifying statements in the current SI. We further note that – as described in the SI of Ref [43] for graphene/Ni we had two independent ways of calibrating the coverage as also added in the revised version of the current SI. We have also added an extensive explanation of the uptake curve behaviour in the SI (starting from the last paragraph on page 1) as indicated by the referee.

In addition to these challenges, to ensure reproducibility and comparable experimental conditions across individual ISF measurements, we performed the dynamical measurements at approximately equivalent levels of specular intensity attenuation I/I_0 and despite applying consistent procedures, some variation in this attenuation is unavoidable; for the 120 K dataset, for example, values range between 0.27 and 0.35.

Accordingly, we have expanded the boundaries of our estimated coverage, which now correspond to a range of approximately 0.12 to 0.20 monolayers.

5) In figure 6a, friction coefficient $\lambda(t)$ are reported, my concern is about the graphene/Ni(111) case that look like its value has not yet reached its asymptotic value at 2 ps.

Author reply:

We thank the reviewer for this comment. We acknowledge that the friction coefficient $\lambda(t)$ for graphene/Ni(111) exhibits a slight increase around 2ps. Our choice of 2 ps as a cutoff was guided by the work of Tocci et. al. (reference 29), which demonstrated that this duration achieves sufficiently converged friction coefficients for reliable estimation. To ensure convergence, we extended our analysis by running additional AIMD simulations with identical parameters and a minimum duration of 3 ps. Here, we observed the friction coefficient rises from $\sim 9.5 \times 10^7 \text{ Nsm}^{-3}$ at 2 ps to $\sim 9.7 \times 10^7 \text{ Nsm}^{-3}$ at 3 ps (an increase of 2 %). The gradient of $\lambda(t)$ remains marginally positive but exhibits significant noise. We do not expect the minimal changes in friction coefficient $\lambda(t)$ to alter our conclusions. Please see section 2.6 in the SI together with Supplementary Figure 13 for the corresponding plot.

We have further added a single error bar to Figure 6(a) and the corresponding reference to the supplement in the main text on page 6:

supported h-BN and graphene. A single error bar is shown for the graphene/Ni system, which illustrates the statistical uncertainty and the degree of convergence when the simulation is extended to 3 ps, further details are provided in Section 2.6 of the SI.

Minor correction:

Main text:

- 1) in "Walking motion of Water on h-BN": " in canonical simulations ((NVT)" remove double parenthesis
- 2) In "Theoretical methods" the unit cell notation " $(\sqrt{7} \times \sqrt{7})R19:1^\circ$ " has to be corrected to " $(\sqrt{7} \times \sqrt{7})R19.1$ "
- 3) the meaning of " periodic images were eliminated. " is not very clear.

In SI

- 1) 1.2 Coverage calibration, line 39 "In Supplementary Supplementary Figure 1", remove repetition
 - 2) page 2, line 76: "v is the preexponential or frequency factor," v is missing in the equation
 - 3) page 4, line 165 & line 178 "hBN"-> "h-BN"
- In conclusion, the manuscript require changes before to be accepted for publication.

Author reply:

Have all been corrected accordingly

Reviewer #3 (Remarks to the Author):

The authors combine helium spin-echo spectroscopy and *ab initio* calculations to show that single water molecules on h-BN/Ni(111) diffuse via a coupled rotational-translational “walking” mechanism—with an activation energy roughly $2.5\times$ lower than the ~ 60 meV barrier on graphene/Ni—despite exhibiting similar binding energies on both substrates. They further demonstrate that metal support dramatically alters friction, with h-BN/Ni displaying markedly lower water friction than graphene/Ni, challenging classical jump-between-site models and underscoring the critical role of substrate polarity in designing 2D-material-based microfluidic systems.

The work is very interesting, it addresses the lack of data for the H₂O on h-BN/Ni system, and it shows marked differences between h-BN and its graphene counterpart case. It is shown that despite the similarity of the substrate and H₂O adsorption energy on h-BN/Ni and graphene/Ni, both experiments and theory show completely different motions of single water molecules on the two surfaces. However, there are a couple comments that need to be addressed before publication.

Author reply:

We thank the referee for their positive evaluation and for recognising the significance of our findings regarding the contrasting behaviour of water on h-BN/Ni and graphene/Ni. We appreciate the thoughtful comments and have carefully addressed the points raised below.

1. What would be the role of entropy in the energetics described in this work? Is entropy accounted for?

Author reply:

We thank the reviewer for raising this important point. Our study concentrates on the dynamics of isolated water molecules adsorbed on h-BN and graphene/Ni(111) surfaces, in the absence of chemical reactions or cooperative processes such as self-assembly. This stands in contrast to systems where configurational entropy or collective interactions play a dominant role in modulating energetics.^{3,4} For single-molecule motion in such weakly bound systems, the relative contributions from vibrational, translational, and rotational entropy are expected to vary minimally across different adsorption sites and during lateral diffusion. Consequently, the entropic term $T\Delta S$ is anticipated to be small relative to the enthalpic component ΔH and unlikely to meaningfully affect the observed energetic hierarchy [see Ref 55].

We also acknowledge the computational constraints inherent in accurately estimating ΔS along the entire simulation trajectory. Methods such as thermodynamic integration, vibrational mode analysis, or harmonic approximations - while valuable - are computationally expensive, especially when applied to systems with high configurational dimensionality or when aiming for spatially resolved entropic mapping. For these reasons, and in alignment with common practice in AIMD studies of similar scope, we have not included explicit entropic corrections at each simulation point.

Nonetheless, we emphasize that our *ab initio* molecular dynamics (AIMD) simulations naturally incorporate thermal fluctuations and permit access to an effective free energy surface under experimental temperature conditions. This approach captures entropic effects implicitly

³ <https://doi.org/10.1021/ja3080117>

⁴ <https://doi.org/10.1021/acs.jpcc.3c06816>

to the extent that they modulate the system's thermal behaviour over accessible time and length scales.

In future work, it would be interesting to explore entropic contributions more explicitly through thermodynamic integration and harmonic approximation for selected configurations to assess their quantitative influence on site preference and diffusion barriers^[65]

We have therefore added the following sentence in the conclusion/outlook section on page 8: for simple metal surfaces. Future studies may also benefit from an explicit quantification of entropic effects on water diffusion, using thermodynamic integration or harmonic analysis, to complement the enthalpic perspective and fully characterise the free energy landscape.^{55,75} Finally, to determine if the observed differ-

2. DFT methods used here are known to overestimate adsorption energies, especially compared to coupled-cluster/QMC methods. How would this overestimation affect calculated diffusion barriers?

Author reply:

We appreciate the reviewer's insightful comment. It is indeed well known that standard DFT functionals, including those used in this study, can overestimate absolute adsorption energies relative to high-level wavefunction-based methods such as coupled-cluster or quantum Monte Carlo (QMC). However, these methods remain largely impractical for extended periodic systems involving surfaces due to their high computational cost and scaling with system size, especially when modelling surface-adsorbate interactions with full geometry relaxation and performing extensive AIMS simulations.

In our case, the diffusion barriers of interest are derived from *relative* energy differences along the potential energy surface (PES), rather than absolute adsorption energies. These relative barriers are typically less sensitive to systematic errors in the underlying exchange-correlation functional, especially when the nature of bonding and local electronic structure does not vary drastically along the diffusion path. This is supported by the excellent agreement we observe between our calculated diffusion barriers and the experimental values obtained via HeSE, as further demonstrated by previous work on weakly adsorbed systems (Refs [43,53,56,88,89,91] in the revised manuscript, plus we have also added some references to benchmarking / accuracy for weakly adsorbed systems with Refs [87,90] in the revised manuscript).

Thus, while we acknowledge that DFT-based methods may not yield quantitatively exact adsorption energies, we believe the relative energetics governing diffusion are sufficiently accurate for the present study.

3. The calculated vibrational spectra for the OH stretching is in the order of 3900 cm⁻¹. This is much higher than normally found for experimental OH stretches (typically below 3750 cm⁻¹). Why is this calculated frequency much higher than experimentally found?

Author reply:

We thank the reviewer for this pertinent observation. We have now clarified this point explicitly in the revised manuscript and the Supplementary Information (Section 2.6, see Refs. [64,65]). As shown in Fig. 6(b–c) of the main text and detailed in the Supplementary Figure S6, the OH stretching modes appear near 3900 cm⁻¹, consistent across both interfacial and isolated water.

To verify the origin of this elevated frequency, we computed the vibrational power spectrum of an isolated water molecule in vacuum. The stretching frequency remains centred near 3900 cm^{-1} , indicating that the observed redshift is not primarily attributable to molecule–surface interactions but rather reflects known limitations of DFT-based harmonic vibrational analyses. Specifically, the overestimation of OH stretching frequencies by 8–12% is a recognised issue arising from the neglect of anharmonic effects and the limitations of commonly used exchange–correlation functionals. Our findings align with benchmark studies – mentioned as Ref 64 and 65 in the revised manuscript – i.e. Medel and Suhm (*Phys. Chem. Chem. Phys.*, **2021**, 23, 5629–5643) and Varandas (*J. Chem. Phys.*, **2022**, 157, 174110), which report comparable overestimations of OH stretching frequencies and advocate for the application of empirical scaling factors to improve agreement with experimental data.

Importantly, our analysis centres on relative frequency shifts - such as the comparison between water on h-BN/Ni(111) and on free-standing h-BN - where systematic errors are expected to largely cancel. This renders the interpretation of substrate-induced effects more reliable, despite the inherent limitations in absolute frequency prediction.

In the main text we refer now to the SI via the following text on page 7:

$\sim 1900\text{ cm}^{-1}$) - see 2.7 in the SI for the assignment and accuracy of the vibrational states.^{64,65} For h-BN, the frequencies of the vi-

While in the SI in 2.7 we have added a new paragraph which explains the mentioned redshift and the recognised issue from DFT in calculating absolute frequencies.

4. In the following sentence in page 2, “A previously reported HeSE study of water motion on graphene illustrated the stark contrast between water diffusion on 2D materials such as graphene and metal substrates, where water is bound much stronger and further influenced by hydrogen bonding between the molecules.” It is not clear in the sentence to which case it refers that water is bound much stronger. Does this refer to the latter (metal surface)?

Author reply:

We have changed the text to: ... weakly interacting 2D materials such as graphene and metal substrates, where water is bound much stronger and further influenced by hydrogen bonding between the molecules.^{23,36,43}

Reviewer #4 (Remarks to the Author):

This is a most interesting paper. It combines state-of-the-art experimentation and simulation to probe the nature of the surface water interactions on h-BN and graphene. The present referee is not an expert in the experimental techniques but they appear to have carefully undertaken. The modelling work is of high quality using state of the art techniques and is clearly reported and reproducible. The results are particularly interesting, especially in the contrast between the diffusion mechanisms on the two systems. The proposed mechanism on h-BN is of special interest although I think it could be more clearly explained. The paper is well written and will be of general interest. Publication in Nat Comms is recommended but I suggest two modifications:

Author reply:

We thank the referee for highlighting the broad relevance of our work and the interest in the proposed mechanism on h-BN. We acknowledge the positive assessment of the quality of our modelling and the recommendation for publication. Based on the constructive suggestions we have revised the manuscript accordingly to improve clarity and address the specific points raised.

1. Can we have a clearer account of the rotational- translational dynamic motion of water, with perhaps a good figure.

Author reply:

Due to the number of trajectories and the complexity of the motion we have produced videos of each of the H₂O/h-BN/Ni simulations. See also our reply to Ref #1, Comment 2:

To facilitate a clearer understanding of the dynamic behaviour, we have provided visualisations of several ab initio molecular dynamics (AIMD) trajectories in the form of supplementary videos. These include three additional trajectories that accompany the revised manuscript. All videos (10 in total) are also permanently archived in the data repository cited under the "Data Availability" section, enabling full access to the underlying motion data.

2. Can we remove "paradigm shift" from the title. This is very interesting and novel work but it is not a paradigm shift in understanding water at 2D interfaces

Author reply:

We agree and since this point was also raised by referee 1 we have now removed "paradigm shift" from the title and instead propose a revised title.

Revised Title: "Single-Molecule Water Motion on h-BN and Graphene: Understanding the Behaviour of Water on 2D Material Interfaces"